

# Predicting inmate suicidal behavior with an interpretable ensemble machine learning approach in smart prisons

Khayyam Akhtar[1], Muhammad Usman Yaseen[1], Muhammad Imran[1], Sohaib Bin Altaf Khattak[2] and Moustafa M. Nasralla[2]

[1] COMSATS University Islamabad, Islamabad, Pakistan
[2] Prince Sultan University, Riyadh, Saudi Arabia

## ABSTRACT

The convergence of smart technologies and predictive modelling in prisons presents an exciting opportunity to revolutionize the monitoring of inmate behaviour, allowing for the early detection of signs of distress and the effective mitigation of suicide risks. While machine learning algorithms have been extensively employed in predicting suicidal behaviour, a critical aspect that has often been overlooked is the interoperability of these models. Most of the work done on model interpretations for suicide predictions often limits itself to feature reduction and highlighting important contributing features only. To address this research gap, we used Anchor explanations for creating human-readable statements based on simple rules, which, to our knowledge, have never been used before for suicide prediction models. We also overcome the limitation of anchor explanations, which create weak rules on high-dimensionality datasets, by first reducing data features with the help of SHapley Additive exPlanations (SHAP). We further reduce data features through anchor interpretations for the final ensemble model of XGBoost and random forest. Our results indicate significant improvement when compared with state-of-the-art models, having an accuracy and precision of 98.6% and 98.9%, respectively. The F1-score for the best suicide ideation model appeared to be 96.7%.

## INTRODUCTION

Suicide is a death caused by one's actions or will with the intent of harming themselves (*Centers for Disease Control and Prevention, 2022*). Suicide remains one of the leading causes of death, with an estimated more than 700,000 deaths worldwide, of which about 77% deaths occur in low and middle-income countries (*World Health Organization, 2021*). Suicidal behaviour covers a broad spectrum of signs, including suicidal thoughts, ideations, communicating plans, and finally, attempting suicide. The control, legality, duration, and other characteristics of such manifestations also differ. Such differences are often the result of culture, lifestyle conditions, family ties, gender, medical history, and age. Furthermore, the suffering associated with someone's death extends beyond just that person. It impacts their family and close ones around them, which in turn could possibly induce suicidal behaviour in them, too (*Fonseca-Pedrero & Al-Halabí, 2021)*.

Corresponding author
Moustafa M. Nasralla,
mnasralla@psu.edu.sa

Through studies done in the past, suicide rates have been observed to vary between genders along with the legality of suicide attempts. Generally, for men, the number of suicide attempts is less than that for women, but are far less lethal due to more lethal methods being used by men. Suicidal behaviour and the number of suicide attempts have been observed to be higher in people suffering from mental disorders like depression and schizophrenia or are involved in drug abuse like alcohol (*Fonseca-Pedrero & Al-Halabí, 2021*; *Hegerl, 2022*).

Suicidal behaviour and its causes remain a complex multi-factorial problem of which timely predicting or classifying a person as suicidal, though very important, remains a challenging classification problem. Alongside the advancements in the artificial intelligence domain, utilizing machine learning techniques for classifying and predicting suicidal behaviour, self-harm, and suicidal ideations has been a growing trend in recent years. In the context of smart prison technologies, the integration of machine learning models for predicting suicidal behaviour among inmates represents a crucial and innovative advancement (*Aldhaheri et al., 2022*). While smart prisons have primarily been associated with enhancing security and control, the well-being of inmates has remained under-explored. Smart prisons, equipped with intelligent monitoring systems and the Internet of Things (IoT) infrastructure, can offer a unique opportunity to address the pressing issue of mental health within the prison environment (*Kaun & Stiernstedt, 2020*; *Singh et al., 2021*).

The convergence of smart technologies and predictive modelling in prisons opens up new avenues for monitoring inmate behaviour, detecting signs of distress, and ultimately mitigating suicide risks (*Ul haq et al., 2020*; *Altaf Khattak, Nasralla & Rehman, 2022*). By harnessing real-time data from inmate databases, our research bridges the gap between smart prison technology and mental health concerns. In this work, we will propose an approach employing machine-learning techniques to create human-readable statements and rules for identifying suicidal behaviour. In a world where urban environments are increasingly linked to the development of mental health disorders, the integration of smart healthcare technologies into the prison context becomes even more significant (*Alwakeel et al., 2023*). By aligning with the principles of smart healthcare, our research not only advances suicide prediction within smart prisons but also contributes to the broader discourse on the intersection of technology, mental health, and imprisonment (*Nasralla et al., 2023*).

Furthermore, big data service architectures can allow for scalable and efficient machine learning pipelines for suicide prediction within smart prisons (*Wang et al., 2020*). As well as blockchain-based scheme for secure data sharing for protecting inmate data privacy while enabling the deployment of machine learning models in smart prisons (*Singh et al., 2021*). Finally, the secure electronic medical record authorization systems can offer a secure method for clinicians to access inmate electronic medical records on smart devices, empowering them to make informed decisions regarding suicide prevention (*Chen et al., 2020*). These studies play a vital role in shaping our future efforts for suicide prediction within the unique context of smart prisons.

Despite the fact that there have been a lot of prior studies on model performances, comparisons, and improving classification outcomes (*Jadoon et al., 2023*; *Mujahid et al., 2023*). Such works are often limited to technical improvements instead of explaining the black-box models for clinician decision-making purposes. There has been some progress on interpreting black-box models through SHapley Additive exPlanations (SHAP) (*Lundberg & Lee, 2017*) and local interpretable model agnostics explanations (LIME) (*Ribeiro, Singh & Guestrin, 2016b*), which can calculate and highlight the important features having the most positive contribution towards accurate classifications (*Fonseca-Pedrero & Al-Halabí, 2021*; *Knapič et al., 2021*; *Nordin et al., 2023*). However, such works still lack condition-based rule explanations that will be easy to understand by clinicians and people in general. Most of the past research often utilises a relatively high number of features compared to ours, which leads to lower accuracy and complex models.

We use anchor explanations, which are based on a specific approach that generates local explanations by finding easy-to-understand rules or conditions that hold for a given instance with high certainty in binary classifications (*Ribeiro, Singh & Guestrin, 2018*). The rules are designed to capture the key features while leaving out the rest, leading to explanations that are more concise in human-readable if-then conditions (*Belle & Papantonis, 2021*). Such simple rules are essential for developing model interpretations that support clinicians' decision-making without requiring a deep understanding of the computer domain. We further discuss anchor explanations in "Interpretation & Dimensionality Reduction via Anchor".

However, for datasets having a large number of features, the anchor's local explanations can suffer in discriminating relevant features, leading to subpar performance and possibly exclusion of essential features (*Nordin et al., 2023*). To address this drawback, we first calculated and highlighted the key features that contributed to classifications through SHAP. Since SHAP interpretations are not straightforward if-then statements, we then use anchor explanations for creating human-readable statements. We use an anchor on the datasets having only those important features which were noted through SHAP. By doing so, we are able to both highlight the essential features that support accurate classifications and interpret those features in if-then statements that are understandable by humans, which is crucial for explaining models to non-computer field experts like clinicians.

The dataset used in our article available freely for research purposes is sourced from the Criminal Justice Co-Occurring Disorder Screening Instrument (CODSI) provided by the Inter-university Consortium for Political and Social Research (ICPSR 27963 study by *Sacks, 2011*). The CODSI study incorporates the Texas Christian University Drug Screen (TCUDS) for substance abuse evaluation and deploys three mental disorder screening components (the Global Appraisal of Individual Needs Short Screener version 1 (GSS), Mental Health Screening Fomr (MHSF), and Modified MINI Screen (MMS)), benchmarking them against the Structured Clinical Interview for DSM-IV (SCID). Additionally, the dataset delves into the influence of race on screening results. This rich dataset, comprising information about 353 inmates in 14 U.S. facilities participating in prison-based drug abuse treatment programs, is an invaluable resource for enhancing our understanding of mental and substance use disorder screening within the criminal justice

context. In our research, we explore this dataset, which encompasses a wealth of variables, to develop an explainable model that can assist clinicians in their decision-making processes regarding suicide risk.

## Interpretability methods

System designers and end users can benefit from knowing why machine learning models perform the way they do in a variety of ways, including model selection, feature engineering, the ability to believe and act on predictions, and more user-friendly user interfaces. Thus, interpretability has emerged as a crucial issue in machine learning, and research into interpretable models has experienced a resurgence in attention.

Such interpretable models are valued for their transparency since they are as accurate as non-interpretable ones in some situations. When interpretability is crucial, they may even be favoured even when they are inaccurate. But limiting machine learning to understandable models is frequently a serious drawback. As a result, model-agnostic explanations of machine learning predictions can address the issue of black-box machine learning models and offer crucial flexibility in the choice of models, explanations, and representations, improving debugging, comparison, and interfaces for a variety of users and models (*Ribeiro et al., 2016*).

Interpretability approaches like SHAP (*Lundberg & Lee, 2017*), LIME (*Ribeiro, Singh & Guestrin, 2016a*), and Anchor (*Ribeiro, Singh & Guestrin, 2018*) can help with the explainability of machine learning models. Below is a brief introduction to each approach but we go in detail on SHAP and anchor in "Methodology".

- **SHAP:** It is based on Shapley game theory and aims to provide each feature a contribution score showing how much it contributes to a certain prediction. Simply said, the model's performance when features are added and removed is used to calculate the contribution score, which is then averaged over a permutation of features set. It offers a straightforward and consistent manner to comprehend the significance of specific features in a model's prediction.
- **LIME:** It operates by picking a particular case or prediction and creating a straightforward, locally accurate model that simulates the complicated model's behaviour close to the selected occurrence. By building a clearer, more straightforward surrogate model, it explains why a specific prediction was made.
- **Anchor:** By determining the key features of a prediction, it develops straightforward, human-understandable "if-then" rules. These rules are intended to be precise and concise, making them understandable to non-experts and promoting confidence in the model's decision-making process. The reason behind its name is the fact that it "anchors" or fixes the most crucial features upon which the prediction value is relying on.

## Ethical considerations

The potential benefits of artificial intelligence (AI) for mental health are evident. Compared to conventional methods, it may aid in more accurate patient diagnosis, better

clinical decision-making, and the detection of suicide risk. Large amounts of text can be analysed with the use of AI technology, such as content from microblogs and electronic health records. The World Health Organisation has set high goals for how suicide rates are tracked, suicidal people are identified, and psychological aid is provided (*Mörch, Gupta & Mishara, 2019*).

Despite its potential for good, AI creates a number of ethical concerns, such as the dangers of discrimination, a lack of respect for personal privacy, and a general lack of transparency. These dangers have been researched, and recommendations have been made to reduce them. However, they frequently are not protected by the present legal frameworks. Guidelines frequently lack procedures for enforcing and reinforcing them, which renders them useless. Significant ethical questions are also raised by the use of AI to mental health and suicide prevention. For instance, it is challenging to gain informed permission when utilising large-scale datasets to prevent suicide and AI-based detection systems may as well make classification mistakes (*Mörch, Gupta & Mishara, 2019*).

Companies must implement standardised and responsible systems to arrive at morally defensible and consistent responses since the ethical issues surrounding the use of AI in suicide prevention lack defined and anticipated consequences. However, there are currently no standards for the moral application of AI to either mental health generally or suicide prevention. The absence of rules may be the reason why ethical concerns with the use of big data or artificial intelligence in publications on suicide prevention are rarely discussed. In an effort to strengthen the ethical oversight of AI, few declarations and checklists of principles have been issued such as the EU published suggestions and standards for "trustworthy" AI (*Floridi, 2021*). It serves as a baseline for responsible AI development. These declarations are a positive step forward, however they frequently do not state how to use them in practice (*Mörch, Gupta & Mishara, 2019*).

In this article, we explore various classification models to achieve optimal performance. We delved into the evaluation and comparison of seven different classification models, which are discussed in detail in "Models", aiming to identify the most effective approach for suicide prediction. After a comprehensive analysis, we determined that a combination of Extreme Gradient Boosting (XGBoost; XGB) and random forest (RF), forming an ensemble model, yielded exceptional results and surpassed the individual models while delivering superior predictive power.

Furthermore, we concentrated on improving the analytical process's efficiency in our study. In order to do this, we reduced the processed dataset to just 12 and 19 features using the anchor library that also serves as the basis for the rules that are generated. The full list of these features are listed in "Interpretation & Dimensionality Reduction via Anchor". Notably, no demographic data, like age, gender, ethnicity, *etc.*, is included in these attributes. It is reasonable to say, then, that the final models only take into account aspects related to mental health and life circumstances, and do not make any discrimination based on demographic criteria.

We also discuss and compare the importance of anchor rules alongside SHAP and pairwise correlation with few medical professionals from psychology and psychiatry.

As further explained in "Optimizing Explanatory Rule Generation", we also refactored the original code of a function in anchor library in order to reduce redundancy and improve the execution speed. Our rule generation speed was significantly increased as a result of this optimisation.

Finally we also test our trained ensemble model on National Survey on Drug Use and Health (NSDUH) dataset for the years 2015–2019 from Kaggle in order to assess the generalizability and achieved greater performance without prior training on the same dataset (*Gallamoza, 2021*).

The novelty of this article lies in forming a workflow for successfully utilizing anchor rules on high dimensional dataset, improving its execution time and achieving classification performance without prior training on NSDUH dataset.

The rest of the article is organized as follows. "Literature Review" discusses background knowledge and past work. "Methodology" explains our work and models. "Experimental Setup" describes the performance metrics that we use to evaluate our model performances. "Results & Comparisons" shows our results and comparisons with the state-of-the-art results. "Conclusion & Future Work" draws a comprehensive explanation of our conclusions and future work.

# LITERATURE REVIEW

To provide a solid foundation for the study, this section presents a comprehensive review of the relevant literature on suicide prediction, outlining the key themes, theories, and empirical findings.

## Electronic health records

In this section, we delve into a comprehensive examination of research papers that primarily emphasize approaches for suicide prediction using clinical datasets. These datasets typically consist of electronic health records, which encompass a diverse range of data types, including numerical, categorical, and a combination of both.

In their respective studies, authors *Nordin et al. (2022)* and *Boudreaux et al. (2021)* conducted systematic reviews of machine learning techniques for suicide prediction, focusing on research articles published between January 2016 and September 2021. *Nordin et al. (2022)* found that ensemble methods were frequently employed, with regression, support vector machines, and decision trees also being popular choices. Bayesian and instance-based methods like k-nearest neighbors were used as well, the reviewed papers generally demonstrated moderate levels of accuracy as shown in Table 1. In contrast, *Boudreaux et al. (2021)* discussed the application of machine learning techniques to healthcare datasets for suicide prediction, highlighting the use of support vector machines, random forests, and ensemble methods. They noted the challenge of data imbalance in suicide prediction datasets and the issue of overfitting in models. Additionally, they emphasized the value of natural language processing for converting unstructured clinician notes into structured data, but they did not address methods for model interpretability, feature reduction, or model simplification.

**Table 1 Accuracy results of methods from articles reviewed in the systematic review (*Nordin et al., 2022*).**

| Article References | Method name | Accuracy |
|---|---|---|
| *Nordin et al. (2021)* | NaÏves bayes | 0.82 |
| *Barros et al. (2016)*, *Kirlic et al. (2021)*, *Kim, Lee & Lee (2021)*, *Nordin et al. (2021)*, *van Mens et al. (2020)* | K-nearest neighbors | 0.73–0.89 |
| *Amini et al. (2016)*, *Chen et al. (2020)*, *Horvath et al. (2021)*, *Jung et al. (2019)*, *Oh et al. (2017)*, *Oh et al. (2020)* | Deep neural network | 0.62–0.78 |
| *Edgcomb et al. (2021)* | Classification & regression tree (CART) | 0.80 |
| *Burke et al. (2020)*, *Barros et al. (2016)*, *Horvath et al. (2021)*, *Lin et al. (2020)*, *Oh et al. (2020)*, *van Mens et al. (2020)* | Decision tree | 0.72–0.91 |
| *Amini et al. (2016)*, *Barros et al. (2016)*, *Choi et al. (2018)*, *Hettige et al. (2017)*, *Kessler et al. (2017)*, *Kirlic et al. (2021)*, *Nordin et al. (2021)*, *Oh et al. (2020)*, *Passos et al. (2016)*, *van Mens et al. (2020)* | Support vector machine (SVM) | 0.78–0.84 |
| *Amini et al. (2016)*, *Nordin et al. (2021)* | Logistic regression | 0.64–0.83 |

*Horvath et al. (2021)*, *McMullen et al. (2021)*, *Oh et al. (2017)* test the effect of suicide ideation on accurate predictions, as a majority of suicide attempters do not disclose suicide ideation before making an attempt. In *McMullen et al. (2021)*, the authors examined the impact of suicide ideation on the predictive validity of suicide crisis syndrome (SCS) using various prediction models, revealing that while suicide ideation need not be a prerequisite for diagnosing SCS, it still plays a significant role in the diagnostic framework. Current suicide ideation was the top-performing item on the chi-squared distribution, albeit contributing minimally to predictive metrics with SCS. In *Horvath et al. (2021)*, the authors used prediction models, including gradient boosted trees, random forest, and logistic regression, along with a neural network, to evaluate the influence of borderline personality disorder (BPD) and antisocial personality disorder (APD) in predicting suicide behavior. They found that BPD was a key factor, while APD had limited relevance, and achieved promising F1 accuracy scores in suicide prediction even without suicide ideation features, with gradient tree boosting excelling. The authors also simplified the feature set for practical application without sacrificing performance. Furthermore, in *Oh et al. (2017)*, a neural network model with 41 input variables emphasized participant-reported subjective symptoms and psychological states over conventional suicide predictors. The model effectively distinguished between suicide and non-suicide attempters, with the Scale for Suicide Ideation (SSI) ranking relatively low among all variables. The study shed light on the importance of various feature importance metrics like SHAP values and cross-entropy values, particularly in the context of a larger feature set.

*Hettige et al. (2017)*, *Kirlic et al. (2021)*, *Navarro et al. (2021)*, *Nordin et al. (2023)*, *Van Vuuren et al. (2021)*, *van Mens et al. (2020)*, *Walsh, Ribeiro & Franklin (2017)* have explored the application of machine learning models, including random forest and Gradient Boosting, in predicting suicide behavior. *Nordin et al. (2023)*, applied random forest and Gradient Boosting to a clinical dataset, with Gradient Boosting outperforming random forest and identifying ethnicity, suicidal thinking, and prior suicide attempts as critical variables. *Navarro et al. (2021)* used random forest on a large population-based

dataset, separating predictions by gender and achieving a 0.5 sensitivity and 0.76 specificity for females, using mean decrease in prediction accuracy to assess feature importance. An ensemble model with various base learners was created for college student predictions (*Kirlic et al., 2021*), emphasizing factors like depression and anxiety. Although feature importance was assessed using different metrics, it was noted that feature interactions were not considered comprehensively. Random forest was also applied to electronic health record data (*van Mens et al., 2020*), where Shapley values were recommended as a more comprehensive alternative to permutation importance. Finally, in a study involving schizophrenia patients (*Hettige et al., 2017*), various machine learning models were compared, with least absolute shrinkage and selection operator (LASSO) showing the best performance, and feature importance interpretations were done using support vector classification (SVC) and Elastic Net, although SHAP was suggested for a more comprehensive understanding of feature importance.

In *Barak-Corren et al. (2017)*, the authors used a naïve Bayesian classifier model on Electronic Health Records (EHR) data, finding that suicidal behavior was more common in men, associated with "separated" marital status, and prevalent among African American and Hispanic patients. The model detected 44% of male and 46% of female suicidal cases but had limitations due to the assumption of feature independence and less flexibility in handling complex decision boundaries. In study (*Ribeiro et al., 2016*), a meta-analysis of 172 studies using Comprehensive Meta-Analysis and MetaDiSc revealed that prior self-injurious thoughts and behaviors were significant risk factors for suicidal thoughts, attempts, and mortality, with prior suicidal ideation being the strongest predictor. Suicide ideation and prior suicide attempts were the best predictors of suicide death, with non-suicidal self-injury (NSSI) only slightly increasing the odds of an attempt.

## Natural language processing

In this section, we conduct a review of papers that concentrate on the utilization of natural language processing techniques for the purpose of suicide prediction, with a primary focus on textual content sourced from social media platforms and textual datasets.

Natural language processing (NLP) (*Bird, Klein & Loper, 2009*) has shown promising results for detection of suicidal behavior in textual datasets and from electronic health records, as discussed by the authors in *Fonseka, Bhat & Kennedy (2019)*, *Velupillai et al. (2019)*. *Velupillai et al. (2019)* emphasize the potential of natural language processing (NLP) in detecting suicidal behavior in textual data, highlighting its superiority over time-consuming risk assessment tools due to its accuracy, feasibility, and speed, particularly on large datasets. They underscore the adaptability of NLP models to the dynamic nature of suicidal ideation, which is heavily influenced by environmental factors. Additionally, the authors stress the significance of leveraging social media as a rich source of online text related to mental health, as many suicides occur without prior mental health assessment or treatment. In *Fonseka, Bhat & Kennedy (2019)*, the authors discuss NLP's role in interpreting and responding in natural human language, particularly in clinical notes from electronic medical records (EMRs), where it achieves relatively high accuracy in predicting suicidal ideation. Furthermore, NLP's real-time data capture on social media platforms

allows for early identification of suicidality, while conversational agents on these platforms aim to reduce suicidal behavior through positive feedback.

## Comparative analysis

Popular machine learning interpretability strategies include LIME, SHAP, statistical methods, and Anchor rules. Each has advantages and disadvantages when it comes to suicide prediction.

LIME offers local interpretations of model predictions by approximating the behavior of the underlying model to help comprehend suicide risk in specific cases. It is flexible and model-independent, but it is susceptible to interpretation instability due to fluctuations in perturbations and can overlook global trends.

By allocating contributions to each feature, SHAP values provide a consistent method to explain the output of any machine learning model and reveal which individual features affect suicide prediction. Scalability and accuracy may be constrained by their computational expense for large datasets and possible oversimplification of nonlinear relationships.

Statistical techniques provide clear frameworks for analysis of coefficients in suicide risk prediction. Complex nonlinear correlations between predictors may be difficult for statistical models to capture, which could result in oversimplified depictions of the risk factors for suicide. They might not take full use of the predictive capacity of machine learning algorithms, which would restrict their capacity to identify minute but crucial patterns in the data that are pertinent to the prediction of suicide.

Anchor rules offer human-readable and interpretable if-then rules that sufficiently and clearly explain model predictions. These rules can be useful for clinical decision-making in the prevention of suicide. Decision boundaries in the feature space can be transparently identified with their help, providing clear insight into the circumstances in which the model predicts a high or low probability of suicide. For high-dimensional datasets, anchor rules may oversimplify intricate relationships in the data and rule generation may be computationally intensive.

To our knowledge in the domain of suicide prediction models, some studies have incorporated the use of SHAP for quantifying feature importance by considering the contributions of individual features and their interactions, even more fewer are studies aiming to enhance the accuracy of the models by utilizing approach of feature reduction on datasets or creating ensemble models for robust performances. Furthermore within our understanding, ours is the first study to use anchor explanations for suicide prediction by creating simple human-readable rules and conditions. We also bypass weakness of anchor against high dimensional datasets by reducing the data to important features only *via* SHAP.

By incorporating SHAP, Anchor explanations, and ensemble methods, we can advance suicide prediction research by improving interpretability, capturing complex feature interactions, enhancing model performance, and generating more comprehensive insights. These approaches offer valuable contributions to the field and have the potential to

enhance our ability to identify individuals at risk of suicide and provide effective interventions to prevent such tragic outcomes.

# METHODOLOGY

To provide a comprehensive understanding of our approach, this section outlines the steps taken to conduct our work, including details about data preprocessing, data balancing, models, dimensionality reduction, ensemble of models and cross validation approaches.

## Overview

Our research article is driven by four primary objectives, each contributing to a comprehensive exploration of the critical issue of suicide prediction.

First and foremost, we endeavor to develop an effective framework for suicide prediction, harnessing advanced machine learning techniques to create a model that can accurately identify individuals at risk. Additionally, we confront the challenges associated with anchor explanations on high-dimensional datasets. Anchors are pivotal for model interpretability, but their effectiveness can diminish in the face of complex, high-dimensional data. Our research seeks to innovate and adapt anchor explanations to overcome these limitations.

Furthermore, we emphasize the significance of providing transparent and actionable interpretations for model decisions using anchor explanations. In an era where AI and machine learning are increasingly integrated into decision-making processes, ensuring that model outcomes are understandable and trustworthy is of paramount importance.

Lastly, our study delves into the impact of including or excluding features related to suicidal ideation in datasets used for predictive models. Understanding how the presence or absence of such features influences model performance is essential for tailoring predictive systems to real-world applications. In summary, our research encompasses a multifaceted approach, striving to improve suicide prediction, enhance model interpretability, and shed light on the influence of suicidal ideation features on predictive accuracy and utility.

The raw dataset comprises an extensive set of 915 features, including numerical and categorical features, which are converted using One Hot Encoding. To ensure efficient analysis, the dataset is preprocessed into two separate subsets. The first subset contains 2,337 features, including suicide ideation features, while the second subset consists of 2,314 features without any suicide ideation features.

The XGBoost algorithm is employed to train models on each of these preprocessed datasets independently. Subsequently, the SHAP method is utilized to identify the most important contributing features from the XGBoost models. These features are then used to create reduced datasets, which are refined to contain 27 features in both datasets.

The reduced datasets are utilized to re-train XGBoost models, taking advantage of the selected 27 features. To interpret the predictions made by these new models, Anchor explanations are applied, generating interpretable rules for each prediction. These Anchor rules are subsequently analyzed to reduce the datasets even further, resulting in the final

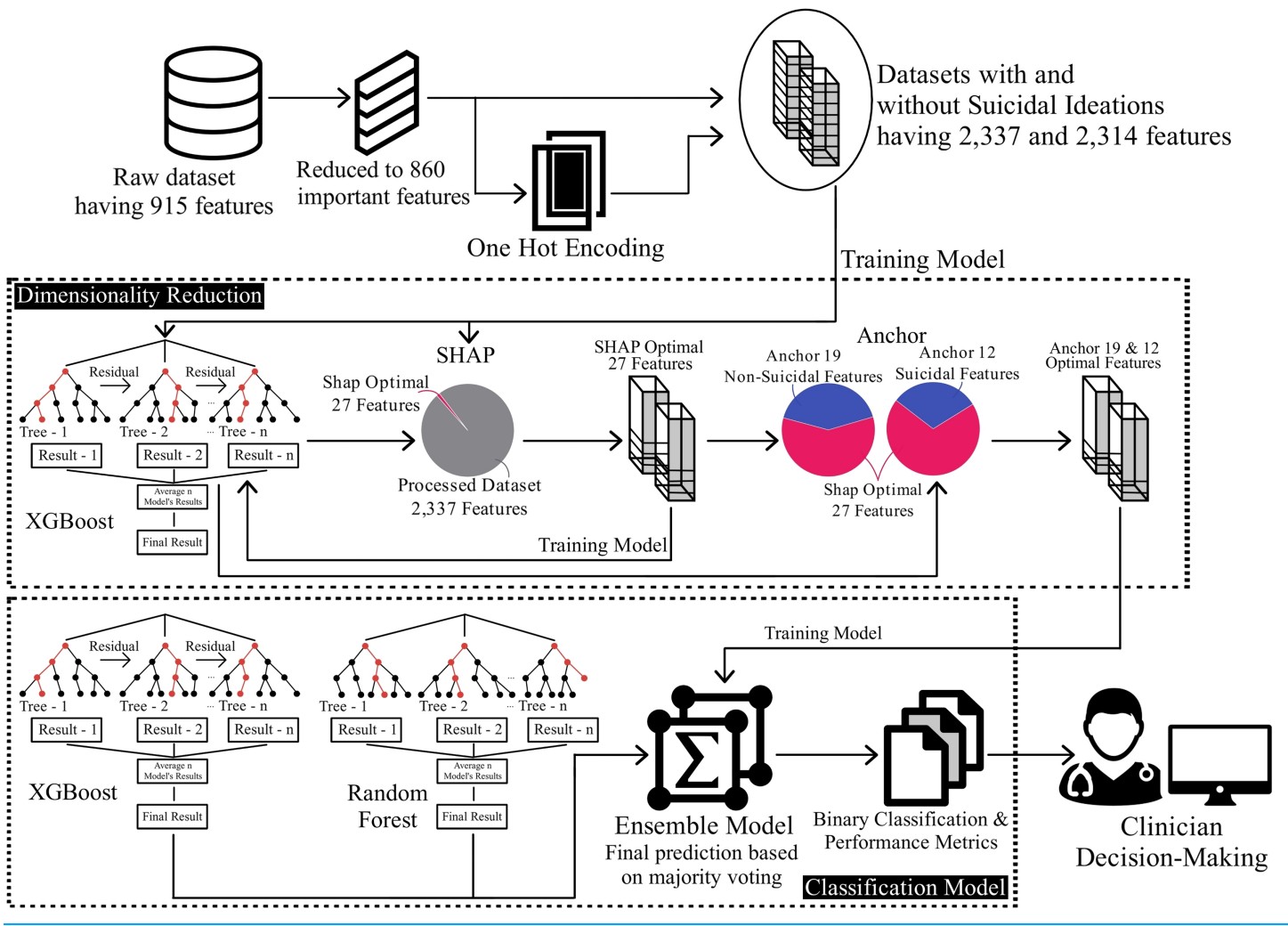

**Figure 1 Overview of methodology for classification on the reduced dataset.**

subsets containing 12 and 19 features, respectively, for including suicide ideation features and without it.

To enhance the predictive performance and exploit the strengths of different algorithms, an ensemble model is constructed by combining XGBoost and random forest. This ensemble model integrates the diverse perspectives and predictive capabilities of both algorithms, leading to improved accuracy and robustness in the final results with the final refined datasets. We also compared ensemble model results with previous datasets, showing fairly improved results with each feature reduction step.

By following this comprehensive methodology, which is also depicted in Fig. 1, this research aims to provide valuable insights into suicide prediction. The combination of preprocessing techniques, feature selection, model interpretation, and ensemble modelling contributes to a holistic approach to understanding and predicting suicide ideation among patients. Each step is discussed in detail in the following subsections below.

| | |
|---|---|
| **Algorithm 1** | **Algorithm to obtain ensemble of XGBoost and random forest models with reduced datasets.** |

1:    (Input: CSV Data File – *D*)

2:    (Outputs: Trained Ensemble Model – *ENS*, SHAP Features – *SF*, Anchor Explanations – *AE*)

3:    $D \leftarrow Load(D)$

4:    $D \leftarrow cleaning(D)$

5:    $D \leftarrow OneHotEncoding(D)$

6:    $D \leftarrow FillMean(D)$

7:    $Suicide\_D \leftarrow D$

8:    $NonSuicide\_D \leftarrow SuicideFiltered(D)$

9:    $DS \leftarrow Suicide\_D, NonSuicide\_D$

10:    **for** datasets $D$ in $DS$ **do**

11:       $T, S \leftarrow 80\_20\_split(D)$

12:       $XGB \leftarrow Fit(T)$

13:       $XGB \leftarrow Pred(S)$

14:       $SF \leftarrow SHAP(XGB)$

15:       $RDS \leftarrow SelectData(SF,D)$

16:    **end for**

17:    **for** datasets $D$ in $RDS$ **do**

18:       $T, S \leftarrow 80\_20\_split(D)$

19:       $XGB \leftarrow Fit(T)$

20:       $XGB \leftarrow Pred(S)$

21:       $AE \leftarrow Anchor(XGB)$

22:       $FDS \leftarrow SelectData(AE,D)$

23:    **end for**

24:    $ENS \leftarrow XGB, RF$

25:    **for** datasets $D$ in $FDS$ **do**

26:       $T, S \leftarrow 80\_20\_split(D)$

27:       $ENS \leftarrow Fit(T)$

28:       $ENS \leftarrow Pred(S)$

29:       $DisplayResults(ENS, AE, SF)$

30:    **end for**

In the Algorithm 1, we represent our process of refining and reducing the dataset for training on an ensemble model consisting of XGboost and random forest. We load the CSV data file (D) and perform data cleaning operations to ensure data quality and remove non-important features. We then apply one-hot encoding to convert categorical variables and fill missing values with mean values. The processed dataset is then split into two datasets: one containing suicide ideation features (Suicide_D) and the other without suicide ideation features (NonSuicide_D).

**Table 2 Feature count in datasets.**

| Dataset description | Total features |
|---|---|
| Raw dataset | 915 |
| Reduced to important features | 860 |
| Categorical features conversion | 2,337 |
| Final suicidal ideation dataset | 2,337 |
| Final without suicidal ideation dataset | 2,314 |

For each of these datasets, we split it into training (T) and testing (S) sets using an 80–20 split ratio. Next, an XGB model is trained on the training set and used to make predictions on the testing set. The SHAP feature values (SF) are calculated using the XGB model, providing insights into the importance of different features in making predictions. We then create two reduced datasets (RDS), each having their top 27 important features only, and proceed to the next stage.

For each of the reduced datasets (RDS), we repeat the training and prediction steps using XGBoost (XGB) and generate anchor explanations (AE) using the XGB model. The anchor explanations provide simple human-interpretable rules that explain the model's predictions per instance/patient. We analyze the anchor rules and further reduce both of the datasets to 12 and 19 features (FDS), including suicide ideation features and without such features, respectively.

Finally, we create an ensemble model (ENS) from XGBoost (XGB) and random forest (RF) models. For each of the final datasets (FDS), the ensemble model is trained and used to make predictions. The results, including the trained ensemble model (ENS), anchor explanations (AE), and SHAP feature values (SF), are displayed.

## Preprocessing

As part of our data processing before training models, we first removed features that were unimportant for model training and prediction, such as fields for dates, location, ID numbers, *etc*. Removing unimportant fields reduced the dataset to 860 features from 915. Even though most of the data for questions were recorded in numerical format and explained through separate directories (*Sacks, 2011*) for the meaning of each numerical data. A few of the questions had additional data fields, namely "other", which was in categorical format. Some examples of the categorical features in the dataset are "Other Specified Current Living Situation", "Other Specified Employment past 6 months", "Specify Other Offense Committed", "Other Specified Drug", *etc*.

For the categorical features, instead of simply mapping categorical values to discrete numerical values, we used Pandas get_dummies (*McKinney, 2011*) one hot encoding to convert it into numerical features, resulting in an increase for the total number of fields to 2,337. Each categorical feature was in its own column, having a value of 1 (exists as a yes) and 0 (exists as a no). This allows for learning the complex combinations of demographics and real-life events per patient that have an impact on suicidal behaviour. The impact of

such demographical and risk factors on suicidal ideation are discussed by authors (*Crepet et al., 1996*; *Goktekin et al., 2018*; *Gould et al., 1996*; *Heikkinen, Aro & Lönnqvist, 1994*; *Nock et al., 2008*; *Zhang et al., 2013*).

We created two separate datasets for training models and comparing. One dataset contained all of the final features, resulting in a total of 2,337 features, this is the dataset with suicidal ideation features included in it. The second dataset was decreased to 2,314 features by removing any feature with suicide terms mentioned in them, this dataset is without suicidal ideation features.

Table 2 shows the total number of feature counts in each dataset.

## Data balancing

For further optimizing the classification performance of each of the seven models, we implemented and compared four Synthetic Minority Oversampling TEchnique (SMOTE) (*Chawla et al., 2002*) variants as well as three NearMiss (*Lemaître, Nogueira & Aridas, 2017*) variants and recorded its results.

SMOTE is an over-sampling technique, but instead of just duplicating the minority class to the equal majority class in terms of data count, the k-nearest neighbour (KNN) (*Kramer & Kramer, 2013*) method is used by SMOTE to generate synthetic data. SMOTE begins by randomly selecting data from the minority class, after which its k-nearest neighbours are determined. The k-nearest neighbour was selected at random, and the random data would then be combined to create synthetic data. This process is repeated until the minority class equals the majority class.

The four variants of SMOTE used are namely SMOTE, Borderline-SMOTE, SVM SMOTE, and Adaptive Synthetic Sampling (ADASYN). Borderline-SMOTE is a variation of SMOTE where, instead of selecting data at random, it exclusively creates synthetic data along the decision boundary of the classes. SVM SMOTE is a variation of Borderline-SMOTE where instead of using KNN, it uses support vector machine (SVM) (*Hearst et al., 1998*) to create synthetic data. ADASYN is a variation of SMOTE where synthetic data is created based on data density (*Chawla et al., 2002*).

All of the SMOTE variants were implemented on the training data features and on the training output variable.

NearMiss is an under-sampling technique where the majority class is reduced to match the minority class size based on the distance between the classes. Three variants of NearMiss were used that are assigned using "*version*" ranging from 1 to 3. In version 1, majority class data is selected, having a minimum average distance from the three closest minority data points. In version 2, majority class data is selected, having a minimum average distance from the three furthest minority data points. In version 3, majority class data is selected, having a minimum average distance from each of the minority data points (*Lemaître, Nogueira & Aridas, 2017*).

All of the NearMiss versions were implemented on the training data features and on the training output variable.

All versions of NearMiss decreased the classification performance of independent models. In contrast, borderline-SMOTE showed better classification performance

compared to other variations of SMOTE. However, during our final reduced datasets and ensemble model, the borderline-SMOTE slightly reduced the classification performance; hence, we do not implement it in the final ensemble model predictions.

## Models

We built total of seven classification models to compare its performance with each other and with results from similar studies. These models were specifically selected due to their reliable performance for handling classification tasks even on small sized datasets.

The seven models were XGBoost (*Chen & Guestrin, 2016*), random forest (*Ho, 1995*), decision tree (*Wu et al., 2008*), a fully connected three layer neural network using Keras on top of TensorFlow (*Chollet, 2015*), logistic regression (*Cox, 1958*), linear regression (*Yan & Su, 2003*) and CatBoost (*Prokhorenkova et al., 2018*). XGBoost and CatBoost are variants of gradient boosting algorithms.

A decision tree is a tree structure resembling a flowchart-like structure, where each internal node represents a test on an attribute, each branch is a test result, and each terminal node (leaf node) is the final class label that will be predicted by a series of tests through the internal nodes (*Wu et al., 2008*).

XGBoost incorporates the use of gradient-boosted decision trees, which are sequentially built. The entire independent variables are given weights and fed into each of the decision trees which predicts results. Each weight of variable predicted wrong by a tree is increased and fed into the next decision tree. These unique predictors are then combined to produce an accurate and robust model. Finding the parameters $\theta$ that suit the training data $x_i$ and labels $y_i$ the best is the essence of training XGBoost, $y_i$ can be from tasks like regression or classification. To determine how well the XGBoost model fits the training set of the data, we must first define the objective function that will be employed. Regularization term and training loss together form the objective function.

$$Obj(\theta) = T(\theta) + R(\theta) \tag{1}$$

where $T(\theta)$ is the training loss function showing how predictive the model is respective to training data, and $R(\theta)$ is the regularization function controlling the model's complexity for avoiding overfitting. Equation (2) illustrates another way to express the objective function.

$$Obj(\theta) = l\left(\sum_k (\widehat{y_k}, y_k)\right) + \left(\sum_r \Omega(f_r)\right) \tag{2}$$

where the training loss function $l(\sum_k (\widehat{y_k}, y_k))$ is showing difference between actual $y_i$ and predicted $\widehat{y_i}$ values, $(\sum_r \Omega(f_r))$ is the regularization term defining complexity of the XGBoost model (*Chen & Guestrin, 2016*).

Random forest is a grouping technique of independent decision tree training forming an ensemble model, having bootstrapping and aggregation as well. Bootstrapping describes the parallel training of many individual decision trees on various subsets of the training dataset using various subsets of the available features. Bootstrapping ensures that every decision tree in the random forest is distinct, which reduces the random forest's total

| Table 3 Hyper-parameters of the models used. | | | | |
|---|---|---|---|---|
| **Models** | **Task** | **Iterations** | **Depth** | **Learning rate** |
| XGBoost | Binary logistic | 100 | 5 | 0.3 |
| Random forest | Gini impurity | 500 | – | × |
| Decision tree | Gini impurity | × | 5 | × |
| CatBoost | – | 100 | 5 | 0.1 |
| Neural network | Adam | 150 | 3 | – |
| Logistic regression | Liblinear | – | × | × |
| Linear regression | × | × | × | × |

**Note:**
   x stands for not applicable, – stands for default parameters.

variance. Through bootstrap aggregation and random feature selection, the predictions are averaged to get the final prediction (*Ho, 1995*).

For XGBoost, the gradient boosting model was used with a learning task set to "*binary: logistic*", which is suitable for binary classification between two classes. The learning rate for XGBoost and CatBoost were set to 0.3 and 0.1, respectively, which controls how rapidly the model changes to adapt to the data. The total number of tree models to create and learn was set to 100, with the maximum depth for each tree set to 5.

For the decision tree model and random forest, *Gini Impurity* was used as criteria for measuring split quality. The total number of trees to generate and train by random forest was set to 500. The maximum depth of the tree was set to five in the case of the decision tree model but was not constrained for the random forest model.

Logistic regression and linear regression were used with their default parameters.

Neural network using Keras was built with an input layer with a number of nodes equal to dataset features size, two hidden layers with node count 12 and eight with activation functions *sigmoid* and *relu* respectively. The final output layer had only 1 node with activation function *sigmoid*. The activation function calculates and decides whether each node should be activated or not. It also gives the model non-linearity, allowing it to adapt to a variety of data and differentiate between the outcomes.

Outputs of linear regression and neural network were probability values ranging from 0 to 1, which we converted to class prediction values with a threshold of 0.5, while the output of other models were discrete class prediction values having values as 0 (non-suicidal) or 1 (suicidal).

The above discussed hyper-parameters are also shown in Table 3.

Each of the seven models was trained separately on both of the datasets (with and without suicidal ideation features), and results were recorded. To further improve the models performance, we used KNN-imputer to fill missing values and Standard-Scalar to standardize the data values by bringing each feature value to unit variance, both methods were implemented through sci-kit learn library (*Pedregosa et al., 2011*) on training and testing of data features.

## Dimensionality reduction *via* SHAP

SHAP is a powerful and versatile model-agnostic method designed to provide local explanations of feature importance in predictive models. It offers a way to decompose a model's predictions into contributions from individual features based on the concept of SHpley values (*Belle & Papantonis, 2021*; *Lundberg & Lee, 2017*; *Shapley, 1953a*).

By leveraging game theory techniques, SHAP enables the interpretation of machine learning model outputs. It employs traditional SHapley values from game theory (*Shapley, 1953b*) and their associated extensions to establish a connection between optimal credit allocation and local explanations.

$$E(b') = \Theta_i + \sum_{j=1}^{K} \Theta_j b'_j \tag{3}$$

where $E$ is representing the explanation model with $b'$ shows the basic features. The maximum size of the collation is represented by $K$, and feature attribution is shown by $\Theta$. Each feature's attribution can be computed by Eqs. (4) and (5) as per the author's recommendation in *Lundberg & Lee (2017)*.

$$\Theta_j = \sum_{S \subseteq I\{j\}} \frac{|S|!(K - |S| - 1)!}{K!} [f_y(S \cup \{j\}) - f_y(S)] \tag{4}$$

$$f_y(S) = X[f(y)|y_S] \tag{5}$$

where a subset of the input features are represented by $S$, a set of all inputs is represented by $I$. Expected value of the function on subset $S$ is represented by $X[f(y)|y_S]$.

A Python package known as Shapash (*MAIF, 2021*) with an aim to make machine learning accessible to all users and interpretable. The project was created by data scientists from MAIF. Shapash is compatible with SHAP and processes local explanations using SHAP's backend.

We used Python packages SHAP and SHAPASH on XGBoost to visualize the important features that provided the most positive contributions to the learning and prediction of models. We were able to reduce both of the datasets having suicidal ideation and without suicidal ideation to only 27 features while retaining the classification performance.

Figure 2 shows the top 27 features that contribute the most to the classification performance of the model without suicidal ideation dataset. Figure 3 shows the top 27 features that contribute the most to the classification performance of the model with the suicidal ideation dataset.

In Figs. 2 and 3, we show a bar plot displaying SHAP contribution values for each data feature, providing a clear visual representation of the impact of individual features on XGboost model's predictions. Each feature is represented by a horizontal bar, where the length of the bar corresponds to the magnitude of its SHAP contribution. Features with higher bars indicate a greater influence on the model's output, while shorter bars suggest relatively lower importance. This visualization allows for a quick and intuitive understanding of which features are driving the model's predictions the most.

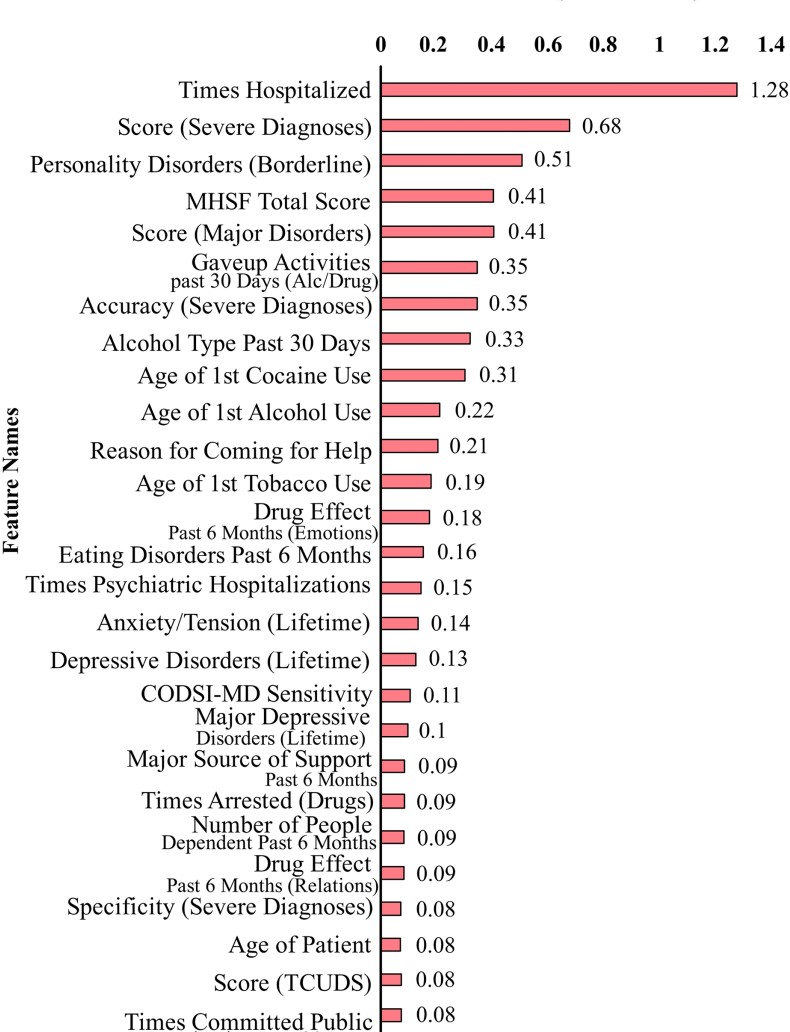

**Figure 2 Feature contributions by SHAP values excluding suicidal ideation features.**

In our study, we use SHAP for identifying and quantifying the impact of individual features on model predictions. The top contributing feature plots for SHAP were used to guide the selection of top 27 features for both datasets. This choice was taken since XGBoost's performance held true for both the original dataset and the smaller dataset of 27 features. This feature selection process is a crucial component of our methodology for aiding in the dimensionality reduction and improving the predictive performance of our models. By strategically employing SHAP to select and prioritize the most influential features, we aim to boost the predictive accuracy and overall performance of our machine learning models, thereby contributing to the advancement of our research objectives.

The algorithm for this subsection is explained in Supplemental Material in greater detail than already shown in Algorithm 1.

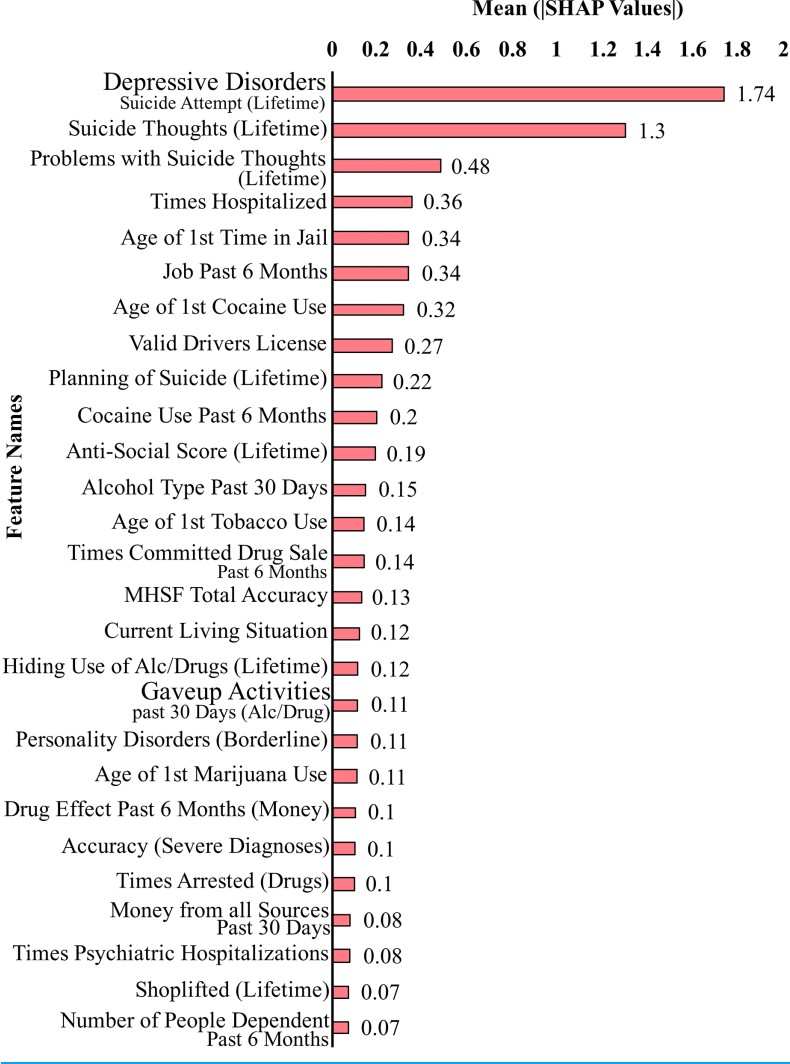

**Figure 3 Feature contributions by SHAP values including suicidal ideation features.**

## Interpretation & dimensionality reduction *via* anchor

In this section, we discuss our approach of anchoring with SHAP-reduced datasets. We also discuss details about anchor explanations and their outputs for our datasets.

Through SHAP, we are able to reduce datasets having suicidal ideations and with no suicidal ideations to its top 27 contributing features while retaining independent model classification performance. We further use anchor explanation (*Ribeiro, Singh & Guestrin, 2018*) on XGBoost to sufficiently anchor each prediction locally. Anchor creates a rule based on perturbation strategy that is much closer to human understandings in such a way that any changes to other features of the instance do not affect the prediction, meaning the prediction on which anchor holds remains almost always the same.

From *Ribeiro, Singh & Guestrin (2018)*, an anchor is formally defined through the Eq. (6):

**Table 4 Some examples of anchor rules from our XGBoost model.**

| # | Performance | Anchor explanation rule |
|---|---|---|
| 6 | Prediction: non-suicidal | (Suicide thoughts lifetime ≤ 0.00) |
|   | Precision: 1 | *AND* |
|   | Coverage: 0.07 | (Number of people dependent past |
|   | Anchor test precision: 1 | 6 months > 4.00) |
|   | Anchor test coverage: 0.01 | |
| 7 | Prediction: suicidal | (Lifetime suicide attempts due to depressive disorders > 0.00) |
|   | Precision: 0.96 | |
|   | Coverage: 0.06 | *AND* |
|   | Anchor test precision: 1 | (Number of hospitalizations due to psych problems in life > 2.00) |
|   | Anchor test coverage: 0.01 | |

$$E_{N_{r(k|P)}}[1_{m(r)=m(k)}] \geq \tau, P(r) = 1 \qquad (6)$$

where $r$ represents an instance that is being explained, basically a row of tabular data. $P$ is a set of predicates meaning the rule or anchor, such that $P(r) = 1$ when all feature predicates specified by $P$ match the feature values of $r$. The classification model XGBoost is represented by $m$. $N_{r(k|P)}$ denotes the distribution of $r's$ neighbours that match $P$. The precision threshold is represented by $\tau$, which in our case was the default parameter of 0.95, only rules with a local fidelity of at least the specified precision threshold are regarded as legitimate outcomes.

We use anchor explanation on both datasets having 27 top contributing features and record prediction rules in a flat file. Through the recorded anchor rules, we were able to observe the most important features for being used as anchors in predictions. We further reduced the dataset having no suicidal ideation features to only 19 features from the previous 27 features and reduced the dataset having suicidal ideation features to only 12 features from the previous 27 features.

Table 4 highlights a few examples, specifically patient 6th and 7th from our data for our XGBoost model, showing the prediction along with the rule that anchors such prediction. Anchor also shows what amount of perturbation space's instances it applies to through its *coverage* value, in which how accurate it is through its *precision* value. For example, in 7th patient prediction. The rule states that since the patient had more than two hospitalizations due to psych problems in the past and has attempted suicide at least once because of depressive disorders, the patient is being classified as suicidal. The rule is 96% accurate in 6% of the perturbation space cases, meaning that the expected outcome is virtually entirely due to the displayed predicates.

Tables S1 and S2 list the important features that anchor explanations selected in if-then statements from the SHAP reduced datasets, along with their total count across entire 71 predictions.

In Figs. S1 and S2, we use grid as a visual representation for depicting pairwise correlations between features in the datasets. Each feature is represented as a label, while

the correlations between the features is represented by colour and value on the grid. The colour of the box provides an approximation of the correlation value, with highly saturated colours like dark blue or near black indicating positive correlations and less saturated colours like white representing negative correlations. By observing the correlation grid, one can quickly identify patterns and relationships between different features, facilitating a deeper understanding of the inter-dependencies within the dataset.

The primary goal of the Anchor library is to produce understandable rules by choosing the features that are most important to the classification. In this instance, the library narrowed down the original set of 27 features to the top 12 and 19 features, respectively. The reason these features were picked in particular is that the Anchor library determined they were necessary for rule generation and omitted the rest by itself, which simplified the model's complexity while maintaining its interpretability.

The algorithm for this subsection is explained in Supplemental Material in Algorithm 1 in greater detain than already shown in Algorithm 1.

## Ensemble of models

The ensemble method is a learning algorithm that combines several machine learning models to form one optimal model, the final predictions are based on weighted votes of predictions from the separate models (*Dietterich, 2000*). We created an ensemble of models using voting classifier (*Pedregosa et al., 2011*). A voting classifier is a machine learning model that predicts a class value by training on an ensemble of several models. It merely averages the results of each classifier that was passed into it to predict the output class based on the highest majority of votes from each classifier. The notion is to build a single model that learns from separate models and predicts output based on their aggregate majority of voting rather than building separate dedicated models and determining the results for each of them.

Initially, we included five models, namely XGBoost, random forest, decision tree, logistic regression and CatBoost, which we reduced to mainly two models, XGBoost and random forest, which gave results similar to the original ensemble. The final ensemble model was trained and tested on reduced datasets from anchor, having 19 features for datasets without suicidal ideation features and having 12 features for dataset with suicidal ideation features.

## Cross validation

In order to rigorously assess the performance of the ensemble models containing XGBoost and random forest, we employed 10-fold Cross Validation and Leave-One-Out (LOO) from *Pedregosa et al. (2011)*. The 10-fold Cross Validation was used on all of the ensemble models recording its accuracy and F1 Score, while Leave-One-Out was used to record accuracy only on all of our models.

These cross-validation techniques are valuable for several reasons. Firstly, the dataset might contain variations and outliers that can affect model performance. By using 10-fold Cross-Validation, the data is divided into ten subsets, and the model is trained and tested on each subset. This provides a robust estimate of model performance and helps to detect

potential overfitting or underfitting. Additionally, LOO, which treats each data point as a separate test case, offers an unbiased evaluation, especially important when working with a relatively small dataset like this. By combining the results from these techniques, we can gain confidence in the ensemble model's generalization capabilities and make informed decisions about its suitability for the specific task with a focus on mitigating overfitting and ensuring reliable predictions.

### Optimizing explanatory rule generation

We modified 'add_names_to_exp' function from 'anchor_tabular.py' from anchor library to remove redundancy and improve execution time of the library in creating explanatory rules by refactoring the original code for execution time efficiency. The Algorithm 2 explains the original workings of the function and Algorithm 3 explains the modified version of the same function.

In the original Algorithm 2 and the modified Algorithm 3, *Indices* contains numerical indices of the feature names from explanation data structures *Exp[Features]*. *Exp[Names]* is an empty list to store names associated with the features when fully processed in to rules. *OrdinalRanges* initializes an empty dictionary to store information about ordinal feature ranges. The first loop over *Indices* iterates and extracts information from the *Mapping* dictionary to determine whether features are ordinal or categorical. If they are ordinal, a maximum and minimum value of ∞ is set to be compared and replaced later on. *HandledSet* is an empty set to keep track of which features have already been handled preventing repeated processing on same feature that is already processed. Second loop over *Indices* iterates for creating final feature explanatory rules. If the operator is *eq* (equal), it constructs a rule based on the feature name and the specific value from the *Mapping* dictionary. Otherwise, for other operators *geq* and *leq*, it calculates ranges and constructs explanatory rules like 'Feature > Value' or 'Value ≤ Feature ≤ Value' based on the ordinal information. Finally, the constructed explanatory rules are appended to the *Exp[Name]* list. In summary, it uses the data structure *Exp* containing numerical processed information of rule generation to process and replace with human-readable feature names based on index mappings and ordinal range information for final outputs from anchor library.

In the modified Algorithm 3, we removed first loop entirely since we can adjust and use *OrdinalRanges* within the second loop, going from two-pass approach to a single-pass approach. This results in only one loop over all *Indices* instead of two loops without any changes in the workings of the code. The modifications results in less execution time since redundant checks and assignments are entirely skipped. In our modified algorithm, we initialize *HandledSet* before the loop and *OrdinalRanges* inside the loop since only *geq* and *leq* operators were being checked. We handle the *eq* operator without considering the *HandledSet* as in original algorithm, and combine *geq* and *leq* handling into a single pass. Both algorithms ultimately create explanatory rules and set results to *Exp[Name]*, but the specific execution order and handling of operators and duplicates differ between the two algorithms.

| Algorithm 2 | Original algorithm to obtain explanatory rules in Anchor Tabular. |
|---|---|

1:    **procedure** ADD NAMES TO EXPLANATIONS(*DataRow*, *Exp*, *Mapping*)

2:        *Indices ← Exp[Features]*

3:        *Create empty Exp[Names]*

4:        *Exp[Features] ← Mapping[Names]*

5:        *Create empty OrdinalRanges*

6:        **for** indices I in Indices **do**

7:           *f, op, v ← Mapping[I]*

8:           **if** *op* is *geq* or *leq* **then**

9:             **if** *f* not in *OrdinalRanges* **then**

10:               *OrdinalRanges[f] ← [−∞, +∞]*

11:             **end if**

12:           **end if**

13:           **if** *op* is *geq* **then**

14:             *OrdinalRanges[f][0] ← MAX(OrdinalRanges[f][0], v)*

15:           **end if**

16:           **if** *op* is *leq* **then**

17:             *OrdinalRanges[f][1] ← MAX(OrdinalRanges[f][1], v)*

18:           **end if**

19:        **end for**

20:        *Create empty HandledSet*

21:        **for** indices I in Indices **do**

22:           *f, op, v ← Mapping[I]*

23:           **if** *op* is *eq* **then**

24:             *creates Fname rule with feature name and categorical value*

25:           **else**

26:             **if** *f* in *HandledSet* **then**

27:               *Continue*

28:             **end if**

29:             *geq, leq ← OrdinalRanges[f]*

30:             *Creates Fname rule with feature names and ordinal values*

31:           **end if**

32:        **end for**

33:        *Exp[Name] ← Fname*

34:    **end procedure**

## Evaluation of generalizability

We obtained the National Survey on Drug Use and Health (NSDUH) dataset for the years 2015–2019 from Kaggle in order to assess the generalizability or transferability of our

| Algorithm 3 | Modified algorithm to obtain explanatory rules in Anchor Tabular. |
|---|---|

1:     **procedure** ADD NAMES TO EXPLANATIONS(*DataRow*, *Exp*, *Mapping*)

2:         *Indices* ← *Exp*[*Features*]

3:         *Create empty Exp*[*Names*]

4:         *Exp*[*Features*] ← *Mapping*[*Names*]

5:         *Create empty OrdinalRanges*

6:         *Create empty HandledSet*

7:         **for** indices I in Indices **do**

8:            *f, op, v* ← *Mapping*[*I*]

9:            **if** *op* is *eq* **then**

10:               *creates Fname rule with feature name and categorical value*

11:           **else**

12:               **if** *f* in *HandledSet* **then**

13:                 *Continue*

14:               **end if**

15:               *OrdinalRanges*[*f*] ← [$-\infty, +\infty$]

16:               **if** *op* is *geq* **then**

17:                 *OrdinalRanges*[*f*][0] ← MAX(*OrdinalRanges*[*f*][0], *v*)

18:               **end if**

19:               **if** *op* is *leq* **then**

20:                 *OrdinalRanges*[*f*][1] ← MAX(*OrdinalRanges*[*f*][1], *v*)

21:               **end if**

22:               *geq, leq* ← *OrdinalRanges*[*f*]

23:               *Creates Fname rule with feature names and ordinal values*

24:            **end if**

25:         **end for**

26:         *Exp*[*Name*] ← *Fname*

27:     **end procedure**

trained ensemble model on a different dataset (*Gallamoza, 2021*). The raw dataset includes 2,812 features for 282,768 participants that were gathered between 2015 and 2019. The features included questions about drug use, thoughts and ideas of suicide, and more.

We searched and compared those features in the NSDUH dataset based on our anchor's top features, which were 12 and 19, respectively, for datasets with and without suicidal ideation features. We were able to test our trained ensemble model on the new dataset by highlighting a total of eight features that overlapped between the two datasets. Table 5 mentions the names of the overlapping features.

Since the NSDUH dataset and the prior dataset are primarily in numerical format, processing of both was done in a similar manner. In order to maintain consistency with the

**Table 5 Common features between the two datasets (*Sacks, 2011*) and (*Gallamoza, 2021*).**

| Feature name |
| --- |
| Age of first cocaine use |
| Lifetime suicide attempts due to depressive disorders |
| Suicide thoughts lifetime |
| Significant problems with suicidal thoughts in life |
| Age of first tobacco use |
| Shoplifting—lifetime |
| Age of first marijuana use |
| Ever attempted suicide |

previous dataset, which had values 0 for no and 1 for yes, we had to first map the values from 2 to 0 for no and 1 to 1 for yes in the NSDUH dataset. Based on information from the guidebooks/codebooks of the dataset, we had to remove records of a small number of participants who had poor feature data or who declined to answer questions. As a result, our final test dataset had eight features for a total of 281,739 participants.

## Validating with medical professionals

In order to validate the importance and understandability of simplistic if-then anchor rules for suicide prediction and clinical scenarios, we discussed our Fig. 3 from SHAP plot, Fig. S2 from features pairwise correlation along with few anchor rules such as shown in Table 4 depicting the same features with several medical professionals in our local area. Generally it has been observed that all of the three samples shown to them were not entirely clear to them at first as they did not have any background from computer science or previously came across such techniques before. Though it has also been observed that upon explaining to them, they were able to understand and grasp the anchor rules the fastest. The pairwise correlation proved to be the most difficult for them to understand or grasp the idea of it. Out of them, four medical professionals from fields of psychology and psychiatry were able to understand and provide valuable feedback which we have discussed below.

In discussions with the medical professionals, it became apparent that there is a consensus regarding the significance of understanding each individual patient rather than relying on analysis techniques such as SHAP or pairwise correlation for entire groups of patients.

Medical professional 1 stressed the significance of taking into account unique patient characteristics, such as the severity of their depression, whether they are contemplating suicide right now, or have attempted suicide in the past. Because suicidal behaviour is influenced by a variety of circumstances and predisposing factors, they emphasised the importance of personalised assessments. Medical professional 2 agrees, pointing out that important elements of personal risk profiles include things like age, gender, social support, and past life experiences.

Although medical professionals 2 and 3 acknowledge the importance of thorough analysis, they recommend giving known or understood key characteristics precedence over minute details. Emotional distress, unemployment, and social standing are among the common themes that they have observed to influence suicide risk in various patient populations. By focusing on the most important risk factors, this method enables a targeted assessment that can help in successful intervention strategies.

Medical professionals express a preference for anchor statements due to their clarity and simplicity in conveying important patient characteristics and risk factors. Unlike complex visualizations like SHAP plots, anchor statements provide clear and concise rules that facilitate understanding and interpretation for both clinicians and stakeholders for each unique patient as an individual. This accessibility enhances the practical utility of predictive models in real-world clinical settings.

## EXPERIMENTAL SETUP

### Dataset

For assessing our models' performance and for the purpose of comparing our findings with those of other papers, we recorded accuracy, F1-score, precision, sensitivity, area under the curve (AUC), positive predictive values (PPV), logloss, true positives, true negatives, false positives, and false negatives. These evaluation metrics are selected to have a well-rounded view of models performance and be able to focus on false negatives along with accuracy.

When comparing two classification models, higher accuracy means that the model is better at correctly identifying both positive and negative cases. An improved overall performance in terms of correctly identifying positive cases is typically regarded as having a higher F1 score. It is a harmonic mean of precision and sensitivity. A higher precision means that the model produces fewer false positive results. A model with lower sensitivity will be less accurate at detecting positive cases. A high AUC shows that the model can simultaneously attain high precision and sensitivity. A higher PPV indicates better performance in terms of accurate positive predictions. The predicted probabilities are more likely to match the actual probabilities when the logloss is lower. In other words, the model assigns larger probabilities to the correct class and is more definite about its predictions.

True positives show that suicidal patients were correctly labelled as suicidal, while true negatives show that non-suicidal patients were correctly labelled as non-suicidal. On the other hand, false positives show how many non-suicidal patients were incorrectly classified as suicidal and false negatives show how many suicidal patients were incorrectly classified as non-suicidal. In this domain, fewer false negatives are more crucial than false positives since we do not want to ignore suicidal patients as non-suicidal.

For this comparative study, dataset was selected from the Criminal Justice Co-Occurring Disorder Screening Instrument (CODSI) ICPSR 27963 study from Inter-university Consortium for Political and Social Research (*Sacks, 2011*). The dataset addresses critical shortcomings in mental and substance use disorder screening tools. These instruments often lack validation for use within the criminal justice system and typically assess only one disorder at a time, limiting their effectiveness in identifying co-

**Table 6  Demographic overview of prisoners.**

| Description | Labels | Total |
|---|---|---|
| Ethnicity | White | 137 |
| | African American | 96 |
| | Latino | 120 |
| Gender | Male | 207 |
| | Female | 146 |
| Marital status | Never married | 142 |
| | Legally married | 83 |
| | Living as married | 31 |
| | Separated | 25 |
| | Divorced | 64 |
| | Widowed | 8 |
| Suicidal thoughts | Yes | 57 |
| | No | 296 |
| Suicide attempts | Yes | 59 |
| | No | 294 |

occurring disorders (COD). To tackle these issues, the CODSI study explores innovative methods to rapidly, accurately, and conveniently evaluate individuals in the criminal justice system for COD. It incorporates the Texas Christian University Drug Screen (TCUDS) for substance abuse assessment and evaluates three mental disorder screening components (GSS, MHSF, and MMS), benchmarking them against the Structured Clinical Interview for DSM-IV (SCID). Additionally, the dataset explores the impact of race on screening results. This dataset is freely available for research, offering valuable insights into enhancing mental and substance use disorder screening within the criminal justice context.

The dataset represents information about US inmates across 14 facilities participating in prison-based drug abuse treatment programs. It contains records for total of 353 prisoners for which we identified 915 independent variables as features and one dependent feature variable namely "suicide ever attempted in life". The dataset is of tabular form mostly consisting of numerical values with few being categorical that we process into numerical values through One-Hot Encoding. Out of the 915 variables in the dataset, some variables were derived using simple formulas (using if-else statements) from other source variables. We included both source and derived variables in our study. Each dataset was divided into 80% training data and 20% testing data using train_test_split (*Pedregosa et al., 2011*).

Table 6 shows the demographics about the prisoners in dataset.

## Platform configuration

In our study and proposed approach, we developed Python notebooks on Kaggle, a web-based data science platform known for hosting datasets, competitions, and facilitating the implementation of machine learning models. We utilized notebooks available through a free Kaggle account, which imposes certain limitations, including a maximum of 30 GB of

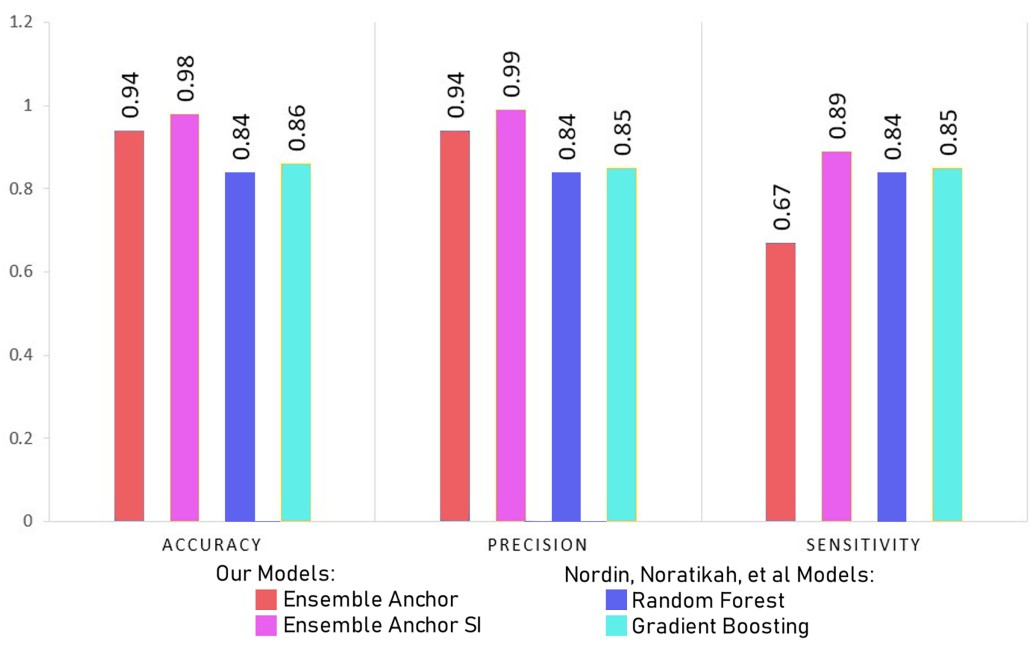

**Figure 4** Comparing model accuracy, precision and sensitivity with *Nordin et al. (2023)*.

RAM access and 73 GB of disk storage. It is worth noting that we refrained from employing any GPU accelerators in our notebooks, models, or methods.

# RESULTS AND COMPARISONS

In this section, we discuss our model results with processed datasets having 2,337 and 2,314 for suicidal ideation (SI) and without suicidal ideation (WSI) features, respectively. We discuss model results with datasets having only 27 features for SI and WSI from SHAP's explanation. We also compare final ensemble results with datasets having only 12 features for SI and 19 features for WSI highlighted from anchor explanations. Finally, we compare our final anchor reduced model performances with state-of-the-art models from *Horvath et al. (2021)* and *Nordin et al. (2023)*.

Models become simpler as the number of features is decreased, which speeds up training and uses less computational power. Furthermore, simpler models are easier to comprehend and interpret, which makes it easier to share the model's insights with stakeholders. Notably, the models' accuracy keeps getting better even with the smaller feature space. This emphasises how crucial feature selection is to improving model performance. The selected features captures the most important parts of the data for classification, leading to better predictive accuracy, even though fewer features could potentially mean losing some information. The interpretability of the models tends to rise with decreasing feature count. It is simpler to comprehend the elements influencing the model's predictions when there are fewer features as discussed in "Interpretation & Dimensionality Reduction via Anchor". Given that stakeholders can more easily understand the reasoning behind the model's decisions, this is in line with the principles of transparency and trust in machine learning models.

**Table 7 Base model performance comparison with processed datasets without feature reductions.**

| Metrics | XGB | XGB[si] | RF | RF[si] | CAT | CAT[si] |
|---|---|---|---|---|---|---|
| Accuracy (%) | 90.140 | 95.770 | 83.100 | 87.320 | 88.730 | 95.770 |
| F-1 Score | 0.830 | 0.927 | 0.651 | 0.749 | 0.570 | 0.888 |
| Precision | 0.901 | 0.958 | 0.831 | 0.873 | 0.887 | 0.958 |
| Sensitivity | 0.643 | 0.786 | 0.286 | 0.429 | 0.111 | 0.667 |
| AUC | 0.804 | 0.893 | 0.625 | 0.706 | 0.556 | 0.833 |
| PPV | 0.818 | 1.000 | 0.667 | 0.857 | 1.000 | 1.000 |
| Logloss | 1.135 | 0.486 | 1.946 | 1.459 | 1.297 | 0.486 |
| True positive | 9 | 11 | 4 | 6 | 1 | 6 |
| True negative | 55 | 57 | 55 | 56 | 62 | 62 |
| False positive | 2 | 0 | 2 | 1 | 0 | 0 |
| False negative | 5 | 3 | 10 | 8 | 8 | 3 |

Note:
XGB, XGBoost; RF, random forest; CAT, CatBoost. Models trained without suicidal ideation features unless shown with SI (with suicidal ideation).

**Table 8 Base model performance comparison with processed datasets without feature reductions.**

| Metrics | NN | DT | LOG | LIN |
|---|---|---|---|---|
| Accuracy (%) | 77.460 | 85.920 | 74.650 | 67.610 |
| F-1 Score | 0.491 | 0.778 | 0.620 | 0.546 |
| Precision | 0.775 | 0.859 | 0.746 | 0.676 |
| Sensitivity | 0.071 | 0.643 | 0.429 | 0.357 |
| AUC | 0.473 | 0.778 | 0.627 | 0.540 |
| PPV | 0.250 | 0.643 | 0.375 | 0.263 |
| Logloss | 0.563 | 1.622 | 2.919 | 2.383 |
| True positive | 1 | 9 | 6 | 5 |
| True negative | 54 | 52 | 47 | 43 |
| False positive | 3 | 5 | 10 | 14 |
| False negative | 13 | 5 | 8 | 9 |

Note:
NN, neural network; DT, decision tree; LOG, logistic regression; LIN, linear regression. Models trained without suicidal ideation features.

The features identified through anchor explanations for WSI and SI are shown through SHAP summary plots as well in Figs. 3 and 4, in the Supplemental Material respectively. These SHAP plots give insights into the driving factors for the model's predictions through color-coded representation for feature value and contributions.

The Tables 7 and 8 show classification performance for our base models trained with datasets of SI and WSI without any feature reductions or SMOTE. The results show better classification performance by XGBoost, CatBoost and random forest, because of which we select XGBoost and random forest for our ensemble model. We did not use CatBoost as part of our ensemble model despite having better performance than random forest since it is generally recommended for datasets having a large number of records, with small datasets, it can have generalizability problems. The Tables 9 and 10 show classification

**Table 9 Base model performance comparison with processed datasets and borderline SMOTE without feature reductions.**

| Metrics | XGB | XGB[si] | RF | RF[si] | CAT | CAT[si] |
|---|---|---|---|---|---|---|
| Accuracy (%) | 90.140 | 92.960 | 91.550 | 88.730 | 88.730 | 80.280 |
| F-1 Score | 0.830 | 0.886 | 0.859 | 0.812 | 0.767 | 0.718 |
| Precision | 0.901 | 0.930 | 0.915 | 0.887 | 0.887 | 0.803 |
| Sensitivity | 0.643 | 0.786 | 0.714 | 0.643 | 0.667 | 1.000 |
| AUC | 0.804 | 0.875 | 0.840 | 0.795 | 0.793 | 0.887 |
| PPV | 0.818 | 0.846 | 0.833 | 0.750 | 0.545 | 0.391 |
| Logloss | 1.135 | 0.811 | 0.973 | 1.297 | 1.297 | 2.270 |
| True positive | 9 | 11 | 10 | 9 | 6 | 9 |
| True negative | 55 | 55 | 55 | 54 | 57 | 48 |
| False positive | 2 | 2 | 2 | 3 | 5 | 14 |
| False negative | 5 | 3 | 4 | 5 | 3 | 0 |

Note:
XGB, XGBoost; RF, random forest; CAT, CatBoost. Models trained without suicidal ideation features unless shown with SI (with suicidal ideation).

**Table 10 Base model performance comparison with processed datasets and borderline SMOTE without feature reductions.**

| Metrics | NN | DT | LOG | LIN |
|---|---|---|---|---|
| Accuracy (%) | 61.970 | 84.510 | 73.240 | 76.060 |
| F-1 Score | 0.523 | 0.748 | 0.608 | 0.632 |
| Precision | 0.620 | 0.845 | 0.732 | 0.761 |
| Sensitivity | 0.429 | 0.571 | 0.429 | 0.429 |
| AUC | 0.593 | 0.742 | 0.618 | 0.653 |
| PPV | 0.240 | 0.615 | 0.353 | 0.400 |
| Logloss | 0.839 | 1.784 | 3.081 | 1.909 |
| True positive | 6 | 8 | 6 | 6 |
| True negative | 38 | 52 | 46 | 48 |
| False positive | 19 | 5 | 11 | 9 |
| False negative | 8 | 6 | 8 | 8 |

Note:
NN, neural network; DT, decision tree; LOG, logistic regression; LIN, linear regression. Models trained without suicidal ideation features.

performance for our base models trained with balanced datasets of SI and WSI through borderline-SMOTE without any feature reduction. The results show better classification performance for linear regression and random forest with borderline-SMOTE, but overall, in most of the cases, even after the ensemble model of XGBoost and random forest, the performance dropped. Hence, we do not use borderline-SMOTE for our final ensemble training and testing.

In Table 11, we are comparing our ensemble model of XGBoost and random forest performance with the above-mentioned datasets. The classification performance metrics show how reducing the number of features in datasets to important contributing features

**Table 11  Model performance comparison with processed datasets, SHAP-reduced datasets, and anchor-reduced datasets.**

| Metric | Ensemble | Ensemble-SI | Ensemble-SHAP | Ensemble-SHAP SI | Ensemble-Anchor | Ensemble-Anchor SI |
|---|---|---|---|---|---|---|
| Accuracy (%) | 91.550 | 95.770 | 92.960 | 98.590 | 94.370 | 98.590 |
| F1-Score | 0.727 | 0.888 | 0.814 | 0.967 | 0.859 | 0.967 |
| Precision | 0.915 | 0.958 | 0.930 | 0.986 | 0.944 | 0.986 |
| Sensitivity | 0.333 | 0.667 | 0.556 | 0.889 | 0.667 | 0.889 |
| AUC | 0.667 | 0.833 | 0.770 | 0.944 | 0.825 | 0.944 |
| PPV | 1.000 | 1.000 | 0.833 | 1.000 | 0.857 | 1.000 |
| Logloss | 0.973 | 0.486 | 0.811 | 0.162 | 0.649 | 0.162 |
| True positive | 3 | 6 | 5 | 8 | 6 | 8 |
| True negative | 62 | 62 | 61 | 62 | 61 | 62 |
| False positive | 0 | 0 | 1 | 0 | 1 | 0 |
| False negative | 6 | 3 | 4 | 1 | 3 | 1 |

**Note:**
SI indicates datasets including suicidal ideation features.

**Table 12  Model performance comparisons with *Nordin et al. (2023)* and *Horvath et al. (2021)*.**

| Metrics | Our Models | | Nordin et al. | | Horvath et al. | | | |
|---|---|---|---|---|---|---|---|---|
| | Ens | Ens-SI | RF | GB | XGB | XGB-SI | RF | RF-SI |
| Accuracy (%) | 94.370 | 98.590 | 84.000 | 86.000 | … | … | … | … |
| Precision | 0.944 | 0.986 | 0.840 | 0.850 | 0.917 | 0.786 | 0.714 | 0.909 |
| Sensitivity | 0.667 | 0.889 | 0.840 | 0.850 | 0.786 | 0.786 | 0.714 | 0.714 |
| AUC | 0.825 | 0.944 | … | … | 0.875 | 0.955 | 0.822 | 0.893 |
| PPV | 0.857 | 1.000 | … | … | 0.786 | 0.786 | 0.714 | 0.714 |
| Logloss | 0.649 | 0.162 | … | … | 0.334 | 0.277 | 1.661 | 1.108 |
| True positives | 6 | 8 | … | … | 11 | 11 | 10 | 10 |
| True negatives | 61 | 62 | … | … | 56 | 54 | 53 | 56 |
| False positives | 1 | 0 | … | … | 3 | 3 | 4 | 4 |
| False negatives | 3 | 1 | … | … | 1 | 3 | 4 | 1 |

**Note:**
SI indicates datasets including suicidal ideation features. Ens indicates our ensemble model trained with anchor features. XGB, RF, GB indicates XGBoost, Random Forest, Gradient Boosting respectively. … indicates metric not available to compare.

by SHAP greatly improves the classification results by reducing the false negatives (suicidal patients that were incorrectly marked as non-suicidal) almost by half in case of WSI and from 3 to only 1 in case of SI dataset. It also shows how further reduction of features through anchor explanations improves the results slightly further, as well in the case of the WSI dataset by improving on the false negative by 1, though there is no change in SI dataset classification. Proving that by using anchor explanations on reduced important features of the dataset from SHAP, not only can we get human-readable if-then statements for explaining the predictions reasoning but also improve classification performance on the further reduced datasets.

In Table 12 and Fig. 4, we are comparing our final ensemble model trained on reduced datasets from anchor explanations with model results from *Nordin et al. (2023)*. The

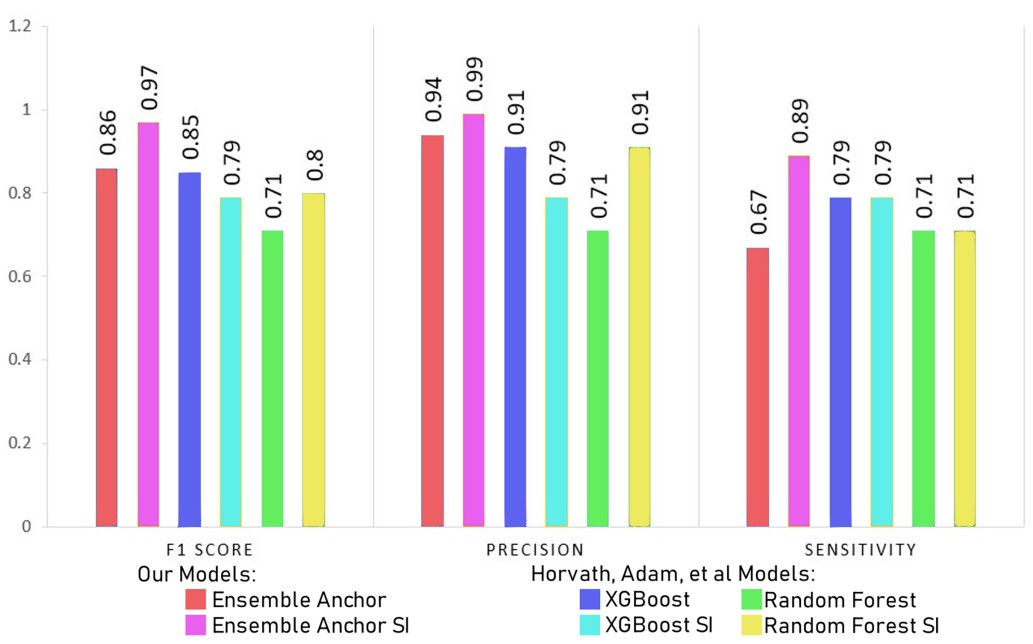

**Figure 5 Comparing model F1-score, precision and sensitivity with *Horvath et al. (2021)*.**

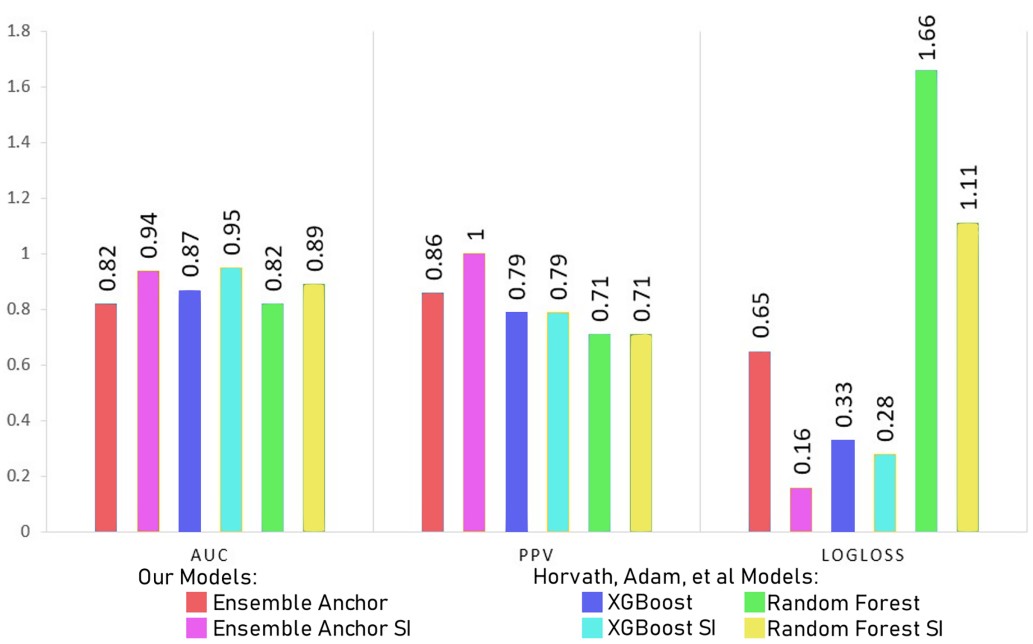

**Figure 6 Comparing model AUC, PPV and log loss with *Horvath et al. (2021)*.**

authors used a different, much smaller dataset than ours, having only 75 patients with 18 total features. Their models are trained on datasets having suicidal ideation features involved in them, for which they highlight past suicide attempts and suicide ideation as the main contributing features. Our ensemble model trained on the SI dataset greatly outperforms the results from their models in terms of accuracy, precision and sensitivity,

**Table 13 Comparison of original ensemble model results with 10-fold cross validation (CV) results.**

| Models | Original accuracy (%) | Original F1-score | CV accuracy (%) | CV F1-score |
|---|---|---|---|---|
| Ensemble | 91.550 | 0.727 | 88.288 | 0.527 |
| Ensemble-SI | 95.770 | 0.888 | 89.729 | 0.567 |
| Ensemble SHAP | 92.960 | 0.814 | 90.394 | 0.669 |
| Ensemble SHAP-SI | 98.590 | 0.967 | 93.990 | 0.796 |
| Ensemble anchor | 94.370 | 0.859 | 90.394 | 0.660 |
| Ensemble anchor-SI | 98.590 | 0.967 | 95.740 | 0.871 |

**Note:**
SI indicates datasets including suicidal ideation features.

meaning our models were able to classify true positives and true negatives more accurately. Compared to the authors' models, our model trained on the WSI dataset slightly lacks only in sensitivity performance, indicating lower performance of correctly predicting positive cases.

In Table 12, Figs. 5 and 6, we are comparing our final ensemble model trained on reduced datasets from anchor explanations with model results from *Horvath et al. (2021)*. The authors used the same dataset but processed it differently as their processed dataset without suicidal ideation has a total of 641 features only, possibly because they removed non-numeric categorical features as well. Our ensemble models outperform theirs except in the case of our WSI-trained model with their XGBoost model, which, too, is trained on a dataset without suicidal ideation features. In this case, our model has slightly lower sensitivity and slightly higher false negatives but lower false positives. Our WSI model still outperforms the other models, and our model trained on the SI dataset outperforms all of the author's models.

The model's predictions in relation to the actual ground truth are summarised in the confusion matrix, as shown in Fig. S5. The confusion matrix in a binary classification scenario, such as ours, is a 2 × 2 matrix. A different combination of predicted and actual class labels is represented by each cell in the matrix. The true positives (TP), true negatives (TN), false positives (FP), and false negatives (FN) components of the confusion matrix are essential for evaluating the predictive power of the model. TP (suicidal predicted as suicidal) and TN (non-suicidal predicted as non-suicidal) are the ones that are predicted correctly, while FP (non-suicidal predicted as suicidal) and FN (suicidal predicted as non-suicidal) are the ones that are predicted incorrectly.

The Table 13 shows the comparison in accuracy and F-1 score of our ensemble models with our original results and with 10-fold cross validation. There is a minor decrease in results compared to our previous results. The Table 14 shows the comparison in accuracy for our original ensemble models with 10-fold cross validation and leave-one-out (LOO). There is a slight increase when using LOO in some cases compared to our cross validated ensemble models.

It is important to recognize that such variations are not uncommon in machine learning experiments. The slight decrease in performance observed with 10-fold Cross-Validation can be attributed to the nature of this technique, which divides the data into ten subsets,

**Table 14 Comparison of 10-fold cross validation (CV) anchor ensemble model results with leave-one-out (LOO) results.**

| Models | Original accuracy (%) | CV accuracy (%) | LOO accuracy (%) |
|---|---|---|---|
| Ensemble | 91.550 | 88.288 | 88.385 |
| Ensemble-SI | 95.770 | 89.729 | 90.652 |
| Ensemble SHAP | 92.960 | 90.394 | 90.085 |
| Ensemble SHAP-SI | 98.590 | 93.990 | 95.751 |
| Ensemble anchor | 94.370 | 90.394 | 90.935 |
| Ensemble anchor-SI | 98.590 | 95.740 | 96.034 |

**Note:**
SI indicates datasets including suicidal ideation features.

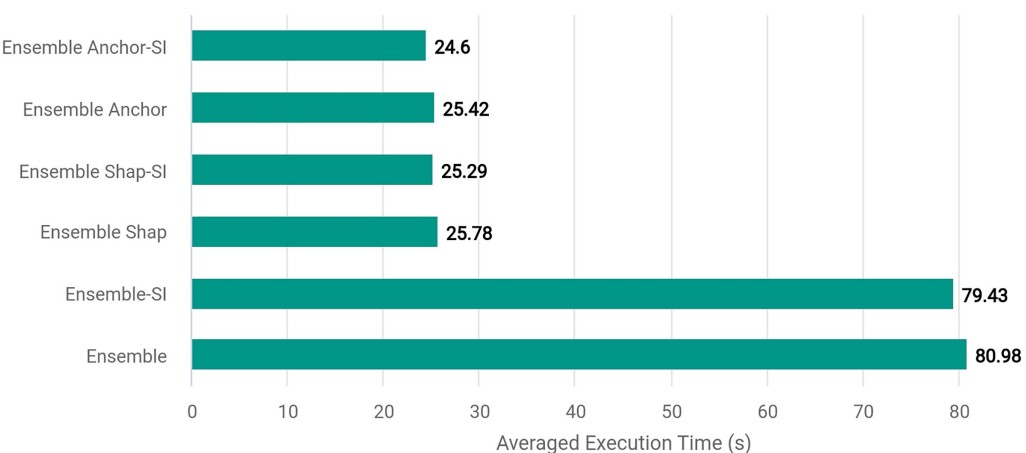

Figure 7 Execution time of our ensemble models on 10-fold cross-validation for performance impact assessment.

potentially leading to a less complex model that generalizes better. On the other hand, the slight improvement observed with LOO suggests that individual data points may contain unique information that contributes positively to the model's overall performance. It is worth noting that the original results, albeit slightly better, still remain within a similar range of performance. Despite various cross validation approaches, our results still show models trained with anchor reduced features to be performing better than entire datasets or SHAP reduced features.

These observations underscore the importance of choosing the most appropriate cross-validation technique for a specific dataset and problem domain. The variations in results can be seen as an opportunity for a deeper exploration of the interplay between model complexity, dataset characteristics, and cross-validation methods, offering valuable insights for future research and model refinement.

The bar plot in Fig. 7 serves as a visual representation of the execution time required for running the ensemble models trained on original, SHAP reduced and anchor reduced

**Figure 8** Comparing execution time of original and modified anchor library across three runs.

datasets with and without SI features while utilizing 10-fold cross-validation procedure. The execution time is averaged for total of three executions of the models. The primary objective being to assess its impact on performance. This analysis seeks to understand the impact of number of features and type of datasets on the training models which results are discussed in Table 13.

## Comparing execution time of anchors

In this section we compare the averaged precision performance and execution time of original anchor library and our modified anchor library for three execution runs of the program. All executions are performed on the top 27 features of SHAP on XGBoost model having same parameters as discussed in "Methodology".

The Fig. 8 compares the execution time for original and modified anchor across three executions separately. The execution time difference between the WSI (Without Suicide Ideation) and SI (Suicide Ideation) models can be observed to be between ranges 50–63% and 44–51% respectively. The Fig. 9 compares the averaged execution time for original and modified anchor across three executions. For the averaged execution times, the difference between the WSI and SI models can be observed to be 45.1% and 38.7% respectively. As per Fig. 9, the total difference in averaged execution time for both WSI and SI models can be observed to be 41.8%.

The Fig. 10 shows the averaged precision value for all 71 patient's explanatory rules for WSI and SI models. We can observe almost no change in the performance across all three execution runs. The Fig. 11 shows the averaged precision value of the three execution runs, where there is no change being observed while having noticeable differences in execution time as shown in Figs. 8 and 9.

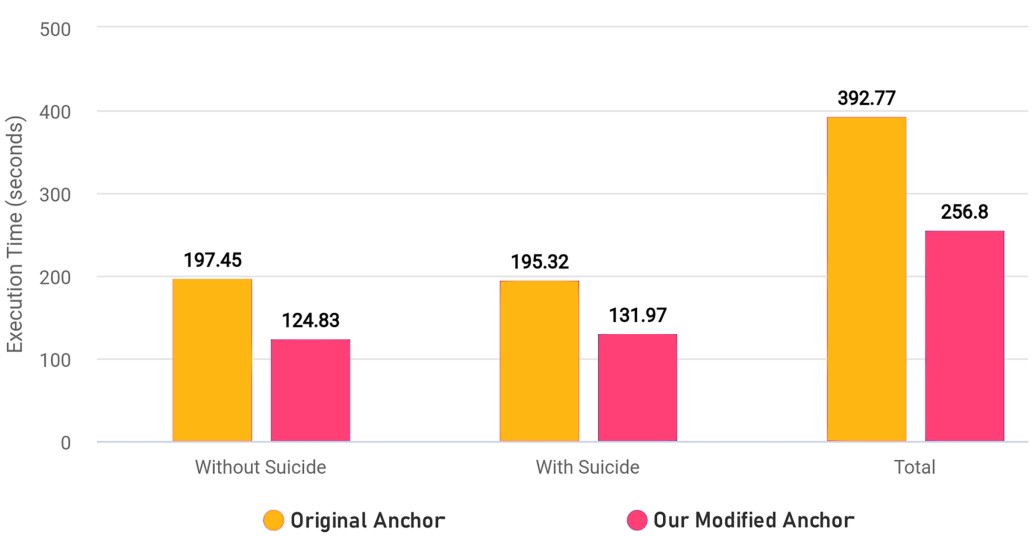

**Figure 9** Comparing average execution time of original and modified anchor library for three runs.

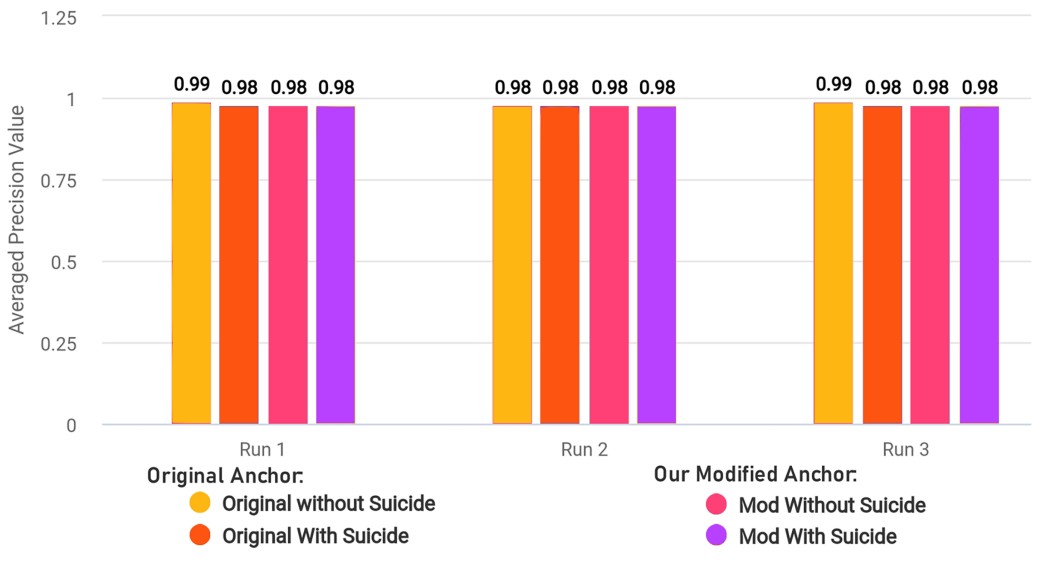

**Figure 10** Comparing average precision of original and modified anchor library for three runs.

## Testing generalizability

In this section we compare the results of our ensemble model on the common eight features with NSDUH dataset discussed in "Methodology".

When the ensemble model was tested on the NSDUH dataset without having been specifically trained on it beforehand, its performance on a number of metrics was noticeably better than that of the original dataset. The accuracy increased significantly

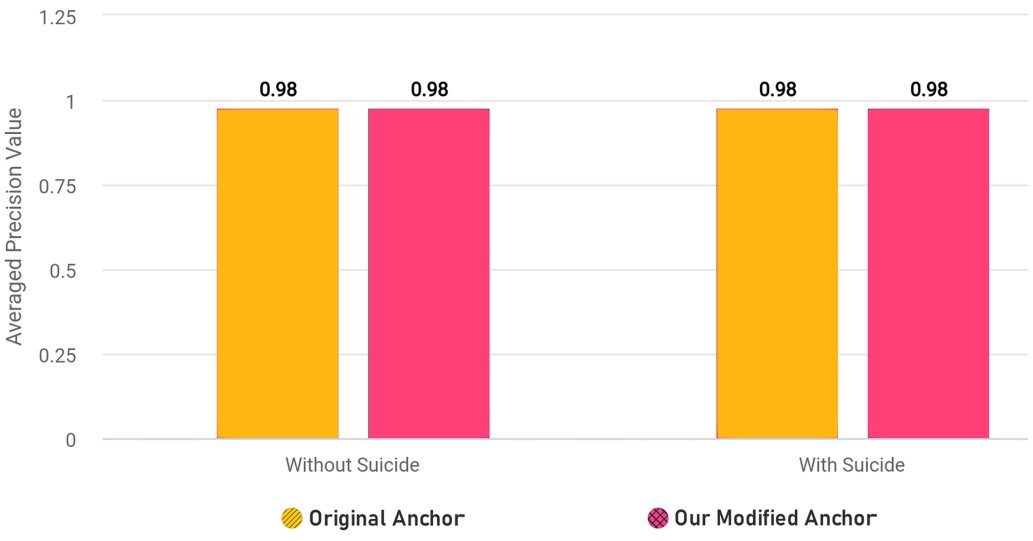

**Figure 11 Comparing average precision of original and modified anchor library averaged for three runs.**

**Table 15 Our ensemble model results trained and tested on *Sacks (2011)* and tested on (*Gallamoza, 2021*) for generalizability based on eight common features.**

| Metrics | Original ensemble | Test on NSDUH dataset |
|---|---|---|
| Accuracy (%) | 95.770 | 99.560 |
| F-1 score | 0.900 | 0.880 |
| Precision | 0.958 | 0.996 |
| Sensitivity | 0.778 | 1.000 |
| AUC | 0.881 | 0.998 |
| PPV | 0.875 | 0.617 |
| Logloss | 0.486 | 0.051 |
| True positive | 7 | 1,813 |
| True negative | 61 | 250,627 |
| False positive | 1 | 1,126 |
| False negative | 2 | 0 |

from 95.770% to a remarkable 99.560%, suggesting a significant improvement in overall predictive capacity. The mentioned results are compared in Table 15. Furthermore, the sensitivity increased dramatically from 0.778 to 1.000, indicating a higher accuracy in identifying positive cases. This significant increase in sensitivity suggests a lower possibility of false negatives, which is important in situations where correctly identifying positive cases is critical.

Additionally, there was a significant increase in the AUC from 0.881 to 0.998, indicating improved ability to distinguish between positive and negative cases. This enhancement highlights the model's resilience and ability to generate more precise forecasts on unknown

data from the NSDUH dataset. Notably, the log loss sharply decreased from 0.486 to 0.051, signifying a substantial decrease in the degree of uncertainty surrounding the model's predictions.

It is interesting to note that although the precision increased slightly from 0.958 to 0.996, the PPV decreased significantly from 0.875 to 0.617. This decrease implies that although the model has improved its accuracy in identifying positive cases, it might now be more cautious when forecasting positive occurrences.

This result demonstrates the model's improved generalizability over a range of datasets and points to its potential for wider real-world application outside of the training set.

## CONCLUSION AND FUTURE WORK

In conclusion, this study presents a novel and explainable model designed to facilitate the generation of human-readable statements, thereby potentially aiding clinicians in their decision-making processes. Our study demonstrates that the inclusion of features related to suicidal ideation significantly enhances the model's performance compared to datasets without such features. Notably, we contribute valuable insights by introducing the use of anchor explanations for predicting suicide behaviour, marking the first instance of employing anchor explanations in this classification context. Furthermore, our research demonstrates a successful strategy for mitigating the limitations of the anchor method through the reduction of dataset dimensions using SHAP before employing the anchor explainer.

In summary, our research underscores the potential of explainable models in assisting clinicians within the smart prison environment and contributes to the advancement of suicide behaviour prediction. As we address the present limitations and encourage further exploration, we anticipate our work to pave the way for more robust and accurate predictive models in the field of mental health within the context of smart prisons and correctional facilities.

Our proposed model's robustness and reliability was tested by validation on a larger dataset with a more extensive sample of patients from NSDUH dataset and achieved greater performance and results without prior training on the same dataset.

The perspectives offered by medical professionals emphasise the significance of customised evaluation in forecasting the likelihood of suicide. Predictive models can assist targeted interventions and improve clinical decision-making by emphasising critical risk factors and concentrating on particular patient characteristics. A viable method for converting intricate predictive models into useful insights that physicians can use to inform tailored patient care is the use of anchor statements. Furthermore, although medical professional 2 emphasized the need to include gender and age in analysis, it should be noted that both of these features were bypassed by SHAP and anchor as other features were contributing more towards the prediction. Also to avoid biasing in the models, these characteristics are best to be left out of final classification features.

However, it is important to acknowledge certain notable limitations in our research.

Firstly, our dataset lacked specific information regarding the type of occupation held by each patient. This information could be particularly pertinent in the context of suicide risk

classification, as the nature of one's job can shed light on their lifestyle, stress levels, and emotional well-being. Therefore, future research should aim to incorporate data about the type of job held by patients, enabling a more nuanced analysis of their mental health and risk factors. It was also highlighted by medical professional 2 and 3.

By addressing these limitations and expanding the scope of our research to encompass a wider variety of patient profiles, we can further enhance the utility and relevance of our model in clinical decision-making processes, making it a valuable tool for mental health professionals in diverse healthcare settings.

## LIST OF SYMBOLS

| | |
|---|---|
| $\theta$ | Parameters to a function |
| $obj\ \theta$ | Objective function to XGBoost |
| $\Omega(f_r)$ | Regularization term For XGBoost complexity |
| **APD** | Antisocial personality disorder |
| **BPD** | Borderline personality disorder |
| **D** | Dataframe of a dataset |
| $E(b')$ | Explanation model |
| $b'$ | Basic features of the model |
| **EHR** | Electronic health records |
| **ENS** | Ensemble model |
| **FDS** | Dataset after feature reduction process |
| $f_y(S)$ | Expected value of a function on subset S |
| **LIME** | Local interpretable model agnostics explanations |
| **NSSI** | Non-suicidal self-injury |
| **NonSuicide_D** | Dataframe of dataset containing suicidal ideation features |
| **NPV** | Negative predictive value |
| $R(\theta)$ | Regularization function |
| **RF** | Random forest classification model |
| **SI** | Suicidal ideations |
| **SMOTE** | Synthetic minority oversampling technique |
| **SHAP** | Shapley additive explanations |
| **SCS** | Suicide crisis syndrome |
| **SITB** | Self-injurious thoughts and behaviors |
| **Suicide_D** | Dataframe of dataset containing SI Features |
| **SF** | SHAP list of top features |
| $T(\theta)$ | Training loss function |
| **T, S** | Training and testing sets of data |
| **T** | Total usage count of feature in AE in all predictions |
| $T^{NS}$ | Total usage count of feature in AE in non-suicidal predictions |
| $T^S$ | Total usage count of feature in AE in suicidal predictions |
| **PPV** | Positive predictive value |

| *XGB* | XGBoost classification model |
|---|---|
| **y** | Actual values |
| $\hat{y}$ | Predicted values |
| **y_i** | Labels for training data |

## ACKNOWLEDGEMENTS

The authors thank the Smart Systems Engineering Lab for providing the needed environment and resources to conduct the research.

### Funding

The authors received support from the Prince Sultan University for the Article Processing Charges (APC) of this publication. The funders had no role in study design, data collection and analysis, decision to publish, or preparation of the manuscript.

### Grant Disclosures

The following grant information was disclosed by the authors:
Prince Sultan University for the Article Processing Charges (APC).

### Competing Interests

The authors declare that they have no competing interests.

### Author Contributions

- Khayyam Akhtar conceived and designed the experiments, performed the experiments, analyzed the data, performed the computation work, prepared figures and/or tables, authored or reviewed drafts of the article, and approved the final draft.
- Muhammad Usman Yaseen conceived and designed the experiments, performed the experiments, performed the computation work, prepared figures and/or tables, authored or reviewed drafts of the article, and approved the final draft.
- Muhammad Imran performed the experiments, analyzed the data, prepared figures and/or tables, authored or reviewed drafts of the article, and approved the final draft.
- Sohaib Bin Altaf Khattak performed the experiments, analyzed the data, prepared figures and/or tables, authored or reviewed drafts of the article, and approved the final draft.
- Moustafa M. Nasralla conceived and designed the experiments, performed the experiments, analyzed the data, performed the computation work, prepared figures and/or tables, authored or reviewed drafts of the article, and approved the final draft.

### Data Availability

The Criminal Justice Drug Abuse Treatment Studies (CJ-DATS) dataset is available in the Supplemental File and at

https://doi.org/10.3886/ICPSR27963.v1.

The dataset was selected from the Criminal Justice Co-Occurring Disorder Screening Instrument (CODSI) ICPSR 27963 study from Inter-university Consortium for Political and Social Research (*Sacks, 2011*).

The NSDUH dataset is available in the Supplemental File and at: https://www.datafiles. samhsa.gov/dataset/national-survey-drug-use-and-health-2015-nsduh-2015-ds0001.

The dataset is available at GitHub and Zenodo:

- https://github.com/KhayyamAkhtar/Inmate-Suicidal-Behavior-Prediction-in-Smart-Prisons/tree/main.

- Khayyam Akhtar. (2024). KhayyamAkhtar/Inmate-Suicidal-Behavior-Prediction-in-Smart-Prisons: Suicidal Behavior Prediction Smart Prisons Source Code v1.0 (v1.0). Zenodo. https://doi.org/10.5281/zenodo.11046490.

## Supplemental Information

Supplemental information for this article can be found online at http://dx.doi.org/10.7717/peerj-cs.2051#supplemental-information.

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
