# Peer review of "Predicting inmate suicidal behavior with an interpretable ensemble machine learning approach in smart prisons"

_PeerJ Computer Science, doi:10.7717/peerj-cs.2051_

## Round 0.1 · original submission · Major Revisions

· Academic Editor

Major Revisions

The manuscript must be revised.

**Language Note:** PeerJ staff have identified that the English language needs to be improved. When you prepare your next revision, please either (i) have a colleague who is proficient in English and familiar with the subject matter review your manuscript, or (ii) contact a professional editing service to review your manuscript. PeerJ can provide language editing services - you can contact us at [email protected] for pricing (be sure to provide your manuscript number and title). – PeerJ Staff

Reviewer 1 ·

Basic reporting

The paper titled "Predicting inmate suicidal behaviour with an interpretable ensemble machine learning approach in smart prisons" focuses on utilizing smart technologies and predictive modelling to revolutionize the monitoring of inmate behaviour and effectively mitigate suicide risks in prison settings. It particularly addresses the overlooked aspect of the interoperability of machine learning models used in predicting suicidal behaviour.

Experimental design

Well defined and explained.

Validity of the findings

Well defined and explained.

Additional comments

The paper presents a promising approach to predicting inmate suicidal behaviour using an interpretable ensemble machine learning approach, addressing the below points that could significantly strengthen the paper.

1. The paper introduces Anchor explanations as a method for creating human-readable statements. However, it's crucial to assess the actual interpretability and usefulness of these explanations in practical scenarios.
2. Detailed case studies or user studies involving domain experts (e.g., prison staff or psychologists) could strengthen the claim of enhanced interpretability.
3. The model's performance and applicability are demonstrated on a specific dataset. However, the generalizability of the model to other prison settings or populations (e.g., different countries or types of prisons) is not discussed.
4. The use of machine learning in sensitive areas like mental health and incarceration raises significant ethical questions, particularly concerning potential biases in the data and the model's predictions. The paper should address how it mitigates biases (e.g., race, gender, socioeconomic status) and the measures taken to ensure ethical use of the technology.
5. While the paper mentions the reduction of features through SHAP and further through Anchor interpretations, it should also discuss the rationale behind the selection of the final features and the complexity of the model.
6. A more detailed discussion on the trade-off between model simplicity, performance, and interpretability would be beneficial.
7. A more comprehensive comparative analysis with existing suicide prediction models and techniques, including a discussion on the advantages and limitations of the proposed approach relative to others, would provide a clearer context for the contribution of this research.

·

Basic reporting

## English language
Generally, the language used in the text is on a good and professional level.
There are abbreviations in the text that are used without first defining them.

## Intro, background and literature
Abstract and introduction clearly state the application area of the research. However, from the introduction of the text it is not obvious what are the computer science achievements that are reported by the authors. Also, it is not obvious what is exactly the nature of the dataset used to verify the results in the text.

In fact, actually, the text represents a comparison between existing methods on the dataset used by the authors. Here the important question is, what exactly is the novelty reported in the text?

Citing within the text contains extra numbers in brackets, for example:
> (Aldhaheri et al., 2022) (5)

Table 1 must also show the information in which work the corresponding method is reported.

## Text structure
The structure of the text is within the standards.

## Figures and tables
The font in the figures is too small which makes the inscriptions hard to read.
Figure 4 and 5 are very good looking, but very hard to understand by the reader.

## Raw data
Authors share Python notebooks that contain the described experiments.

Experimental design

## Originality of the research
The text compares existing methods on a public data set. I am afraid I cannot agree for any novelty and originality claims in the proposed text.

Validity of the findings

## Impact and novelty
I cannot report any novelty in this proposition.

## Underlying data, statistically sound
Data is statistically well-stated as far we are taking about an existing data set, that is not part of the presented work.

Additional comments

The text is just a comparison of the performance of existing Python libraries on a publicly available data set. A similar proposition might be if we take the MNIST dataset and we compare the success of the different Python libraries, applied on the handwritten digits recognition problem.

---

## Round 0.2 · accepted · Accept

· Academic Editor

Accept

Based on the reviewer comments, the manuscript is accepted.

Reviewer 3 ·

Basic reporting

The article has significantly improved in terms of clarity and presentation. The use of professional English is consistent throughout the manuscript, ensuring that the text is clear and unambiguous. However, the following areas require minor adjustments:

All figures and tables are relevant and well-presented.
The raw data supporting the conclusions are shared.

Experimental design

The manuscript aligns well with the journal's Aims and Scope and presents original primary research. The research question is clearly defined and fills a noticeable gap in the knowledge base.

Validity of the findings

The findings of the manuscript are robust, statistically sound, and well-supported by the data. The conclusions are clearly linked to the original research question.

Additional comments

The revised manuscript has successfully addressed all the critical issues previously identified. The enhancements in methodology, statistical rigor, and theoretical integration are particularly noteworthy and have significantly elevated the manuscript’s contribution to the field. The authors have demonstrated a thorough understanding of the subject matter and have responded constructively to enhance the manuscript substantially.

Given the above improvements, I recommend the manuscript be accepted for publication.

·

Basic reporting

I haver reviewed the version submitted by the authors after a round of reviews. The text is clear and the structure of the paper acceptable. References are adequate. The results discussed in the de conclusions seem to be well supported by the data.

Experimental design

I have nothing else to add regarding the experimental design.

Validity of the findings

no comment

Additional comments

The authors have addressed properly the previous concerns of the reviewers.